

# Exploring river–aquifer interactions and hydrological system response using baseflow separation, impulse response modelling and time series analysis in three temperate lowland catchments

Min Lu[1,2], Bart Rogiers[1], Koen Beerten[1], Matej Gedeon[1], Marijke Huysmans[2,3]

[1]Institute for Environment, Health and Safety, Belgian Nuclear Research Centre, Mol, 2400, Belgium
[2]Department of Earth and Environmental Sciences, KU Leuven, Leuven, 3001, Belgium
[3]Department of Hydrology and Hydraulic Engineering, Vrije Universiteit Brussel, Brussels, 1050, Belgium

*Correspondence to*: Min Lu (mlu@sckcen.be)

**Abstract.** Lowland rivers and shallow aquifers are closely coupled and their interactions are crucial for maintaining healthy
stream ecological functions. In order to explore river–aquifer interactions and lowland hydrological system in three Belgian catchments, we apply a combined approach of baseflow separation, impulse response modelling and time series analysis over a 30–year study period at catchment scale. Baseflow from hydrograph separation shows that the three catchments are groundwater-dominated. The recursive digital filter methods generate a smoother baseflow time series than the graphical methods, and yield more reliable results than the graphical ones. Impulse response modelling is applied with a two–step
procedure. The first step where groundwater level response is modelled shows that groundwater level in shallow aquifers reacts fast to the system input, with most of the wells reaching their peak response during the first day. There is an overall trend of faster response time and higher response magnitude in the wet (October–March) than the dry (April–September) periods. The second step of baseflow response modelling shows that the system response is also fast and that simulated baseflow can capture some variations but not the peaks of the separated baseflow time series. The time series analysis indicates that components
such as interflow and overland flow, contribute significantly to stream flow. They are somehow included as part of the separated baseflow, which is likely to be overestimated from hydrograph separation. The impulse response modelling approach from the groundwater flow perspective can be an optional method to estimate the baseflow, since it considers some level of the physical connection between river and aquifer in the subsurface. Further research is recommended to improve the simulation, such as giving more weight to wells close to the river and adding more drainage dynamics to the model input.



## 1 Introduction

In riverine environments, stream flow quantity and water quality are often largely influenced by groundwater via flow and solute exchange. River–aquifer interactions impact the stream ecological functions since a lot of vegetation types are highly dependent on a healthy flow regime and nutrient level. Their interactions, on the other hand, are directly linked with and influenced by changes in climate, land use, land cover, water management policies, and other human activities. Emerging

hydrological stresses, such as observed record droughts in recent decades in Europe, have already resulted in low stream discharge, stream stage and groundwater level (Fu et al., 2020; Hänsel et al., 2019; Spinoni et al., 2017; Laaha et al., 2017). Therefore, understanding river–groundwater interactions and exploring their temporal evolution are of crucial importance, and can provide insights and assistance for future environmental management decisions to minimize the potential adverse effects and maintain a healthy hydro-ecological balance.

Lowland catchments in temperate regions are characterized by flat topographies and shallow groundwater tables. Rivers and groundwater in these catchments are closely coupled and influenced by climatic drivers, such as precipitation (van Walsum et al., 2002). Previous studies conducted on lowland river–aquifer interactions have their distinctive objectives and applied methodologies. For example, to quantify the exchange fluxes, there are hydraulic or heat tracer approaches (Krause et al.,

2012), or event-based hydrochemical and isotopic tracer methods (Poulsen et al., 2015). Process modelling approaches are often used to simulate the river–aquifer interactions. The ubiquitous, and open source, numerical groundwater modelling code MODFLOW has for instance several options for two-way surface water and groundwater interactions (Niswonger and Prudic, 2005; Di Ciacca et al., 2019; Nützmann et al., 2013). More fully integrated models, taking into account the physics of the surface and subsurface domains, such as HydroGeoSphere (Alaghmand et al., 2016), exist as well. Comprehensive reviews on

approaches and applications for better understanding river–groundwater interactions can be found in recent papers and books (Brunner et al., 2017; Cushman and Tartakovsky, 2016; Barthel and Banzhaf, 2015).

In Belgium, research projects with a focus on river–aquifer interactions have been intensively carried out in the Aa river, a typical Flemish lowland river, using a combination of multiple methods, such as heat tracer, river bed hydraulic conductivity

measurements and numerical modelling approaches (Anibas et al., 2009, 2011, 2015, 2017; Ghysels et al., 2018, 2021; Schneidewind et al., 2016). However, these applications are limited to relatively small spatial scale studies, e.g. at the point scale or a short river segment, and difficult to upscale due to the complexity of small-scale heterogeneity in river bed materials and river morphology (Ghysels et al., 2018). Moreover, the research periods in these projects cover relatively short temporal scales, where field data were collected on a few days at chosen seasons (summer and winter) or for a maximum of a few years.


Few studies have assessed the river–aquifer interactions at a larger spatial and temporal scale in Belgian lowland catchments, using methods other than classic baseflow separation techniques (Zomlot et al., 2015; Batelaan and De Smedt, 2007), even





though this seems to be crucial in understanding the connectivity between river water and groundwater. The classic automated baseflow separation techniques can often result in a wide range of outcomes and it is in most cases not very clear which

separation method is superior to the others (Partington et al., 2012). Furthermore, there is little research done in the selected lowland catchments (see Sect. 2) with respect to river–aquifer interaction studies.

In order to fill the research gap, we use a data-driven impulse response modelling approach to quantify the catchment scale hydrological response to meteorological forcing, and the resulting groundwater levels and baseflow, which represents the

river–aquifer interactions, over a 30-year period in three distinctive temperate lowland catchments. Compared to a distributed hydrological model, a lumped model is less time-consuming to construct and can be more effective to transform the input impulse to yield an accurate prediction of the system output such as groundwater level (Long, 2015; Olsthoorn, 2007; Asmuth and Knotters, 2004). Besides the model approach, classic baseflow separation techniques and time series analysis of multiple meteorological and hydrological variables are also applied to assess the groundwater discharge to river and the temporal

evolutions of these variables over the study period.

With the combined approaches above, we aim to

  1. simulate the hydrological process of groundwater level in response to system input of precipitation and air temperature, and the baseflow in response to system input of groundwater level, respectively;

2. estimate the groundwater contribution to stream flows and provide feedback on the application of different baseflow separation techniques by comparing the corresponding results from the impulse response modelling and automated baseflow separation methods; and

  3. investigate the temporal variation, trend and seasonality of the meteorological and hydrological variables that characterize the lowland hydrological system.

**2 Study area**

The study focuses on three temperate lowland catchments in the northeastern and central Belgium: the Zwarte Beek, the Herk and Mombeek (main tributary of the Herk), and the Dijle catchments (Fig. 1). They are sub-catchments of the Scheldt river basin and cover an area of approximately 95, 272 and 893 km$^2$, respectively. The elevation ranges are 21–148 m TAW (Tweede Algemene Waterpassing) for the Zwarte Beek, 25–133 m TAW for the Herk and Mombeek, and 9–177 m TAW for the Dijle.

The highest point in the Zwarte Beek (148 m TAW) is the spoil tip of a coal mine. The climate of the area is humid temperate, with an average annual precipitation ranging from 710 to 820 mm (KMI, 2020) and an average daily air temperature of approximately 11 °C (KMI, 2020), observed between 1990 and 2019 from the nearby meteorological stations (Fig. 1a). There is no strong seasonality for the precipitation; on the contrary, a pronounced seasonality is observed for the air temperature (Fig.





2). The average annual potential open water evaporation varies between 662 and 675 mm, of which the summer potential

evaporation takes up approximately 85 % of the total amount (Batelaan and De Smedt, 2007).

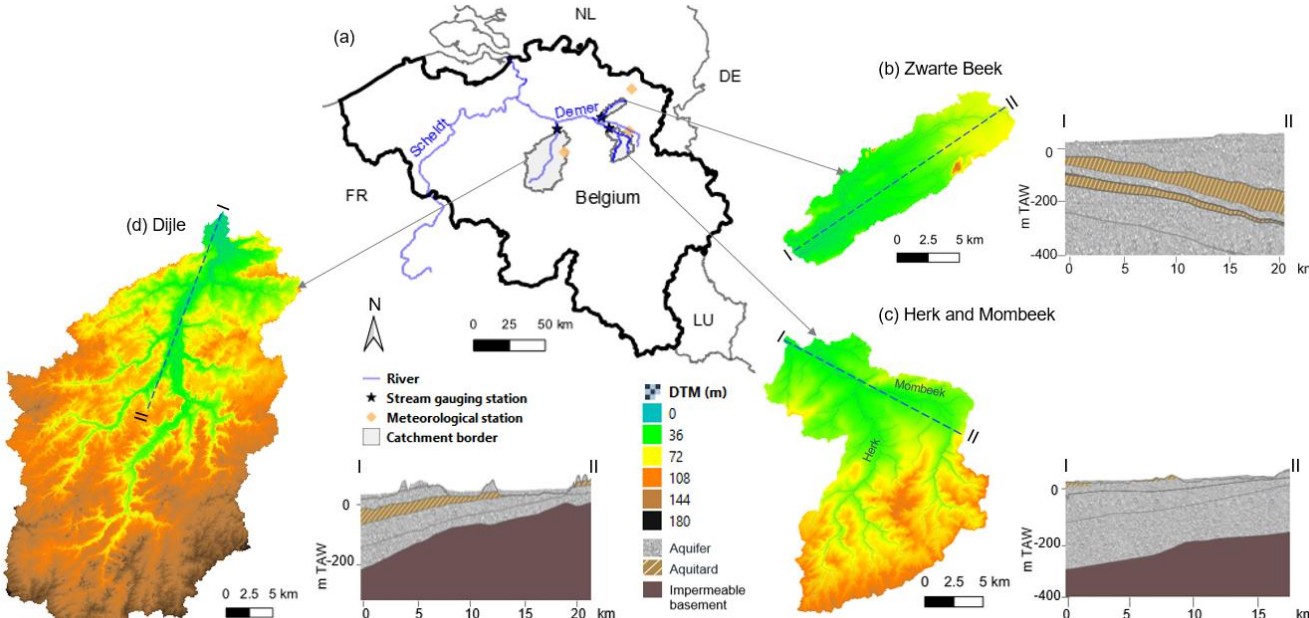

**Figure 1** Location of the study catchments in (a) the Scheldt river basin: (b) the Zwarte Beek, (c) the Herk and Mombeek, and (d) the Dijle; and the cross-sectional sketch of the hydrogeological layers in each catchment (DOV, 2020). DTM from Geopunt Vlaanderen (https://www.geopunt.be/).

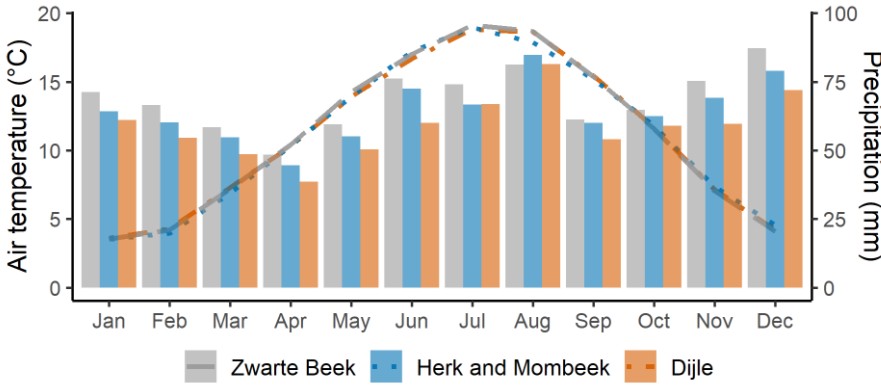


**Figure 2** The average monthly precipitation (bar) and air temperature (line) of the three catchments, based on daily observations between 1990 and 2019 (KMI 2020).

Both the Zwarte Beek and the Herk and Mombeek are tributaries of the Demer (Fig. 1a). The Dijle, the biggest river among

the three, originates in the Walloon region of Belgium. The Demer joins the Dijle at Rotselaar, then heading towards the



Scheldt estuary in Antwerp (Fig. 1a). The stream flows of the studied river segments are measured at the catchment outlet (Fig. 1a) with data obtained via the wateRinfo R package interface (Van Hoey, 2020). The average daily stream flows are 1.05 m$^3$ s$^{-1}$ for the Zwarte Beek, 1.44 m$^3$ s$^{-1}$ for the Herk and Mombeek and 6.65 m$^3$ s$^{-1}$ for the Dijle. The stream flow varies between 0.001–7.03 m$^3$ s$^{-1}$ for the Zwarte Beek, 0.06–20.4 m$^3$ s$^{-1}$ for the Herk and Mombeek, and 1.23–29.1 m$^3$ s$^{-1}$ for the Dijle (Fig.

3a). The flow duration curve (FDC), which plots the cumulative frequency of the stream flow, represents the variability and distribution of stream flow in a catchment (Vogel and Fennessey, 1994), and its shape reflects the catchment's hydro(geo)logical characteristics, with a flat slope indicating a groundwater-feeding surface storage and a steep slope revealing flashy flow regime dominated by direct runoff (Searcy, 1959). The flat slope of the FDCs (Fig. 3a) indicates that these perennial rivers have a strong groundwater feeding feature. The monthly average stream flows show that there is some seasonality in the

time series, specifically, relative higher flows are observed during the winter months in the three catchments (Fig. 3b). For the Dijle catchment alone, the monthly average stream flows are relatively low in the spring and relatively high both in the summer and winter during the 30-year study period (Fig. 3b).

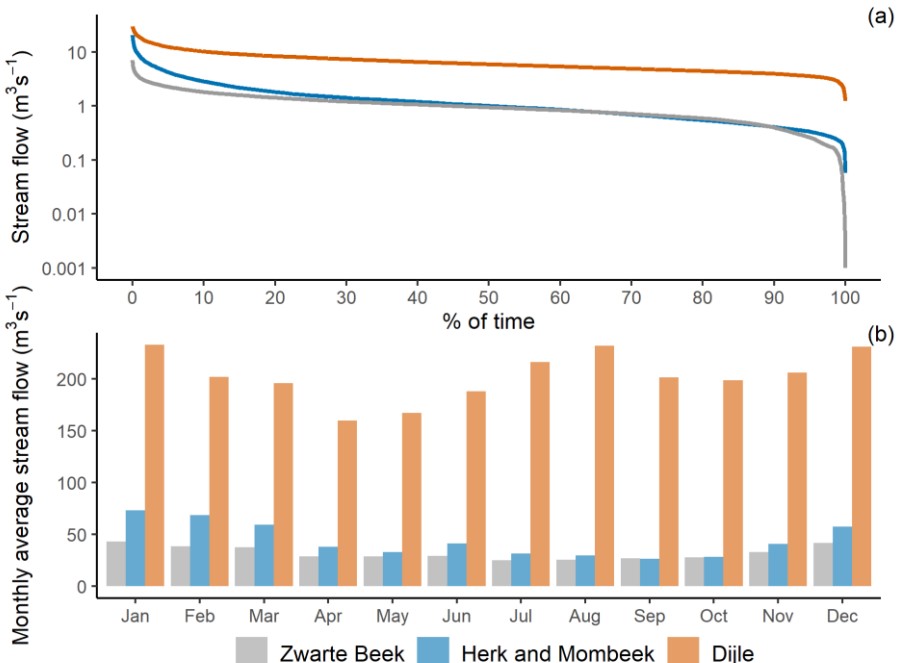

**Figure 3** The FDC (a) and the average monthly stream flow (b) in the three catchments.


Besides different stream flow characteristics, the three study sites have distinctive lithological and hydrostratigraphical settings. The Zwarte Beek valley (Fig. 1b) mainly developed in sandy Neogene sediment that is sitting on top of the Kasterlee clay, which acts as an aquitard (DOV, 2020). The surficial geology outside the floodplain is dominated by sand while the floodplain itself is dominated by sand and peat (DOV, 2020). The sand in the stream bed varies from coarse sand upstream to

fine sand downstream, observed during field trips. The Herk and Mombeek catchment (Fig. 1c) developed mostly in sandy





Palaeogene sediment which lays on top of poorly permeable marl of the Heers Formation and clay from the Hannut Formation (DOV, 2020). The surficial geology outside the floodplain is dominated by loam and sandy loam while the floodplain itself is characterised by loam and clay sediment (DOV, 2020). The river bed of the Mombeek tributary consists mainly of clay as observed during field trips. The Dijle valley (Fig. 1d) mainly developed in Palaeogene and Neogene sands, but continued

incision is the reason why the floodplain has reached the very impermeable clay from the Kortijk Formation below (DOV, 2020). The shallow geology outside the floodplain is dominated by loam and sandy loam deposits (DOV, 2020). The floodplain itself consists of loam, sand, clay and peat (DOV, 2020).

The major land uses are (1) crop (24.4 %), meadow (23.6 %) and urban area (15.2 %) for the Zwarte Beek catchment; (2)

meadow (56.0 %), orchard (21.1 %) and crop (11.3 %) for the Herk and Mombeek; and (3) crop (35.6 %), meadow (28.1 %) and urban area (19.7 %) for the Dijle in 2012 (DOV, 2020). Regarding the water use in Flanders, groundwater and surface water account for 46.4 % and 53.6 % of the tap water supply in 2019, respectively (Vlaamse Milieumaatschappij, 2020).

## 3 Methodology

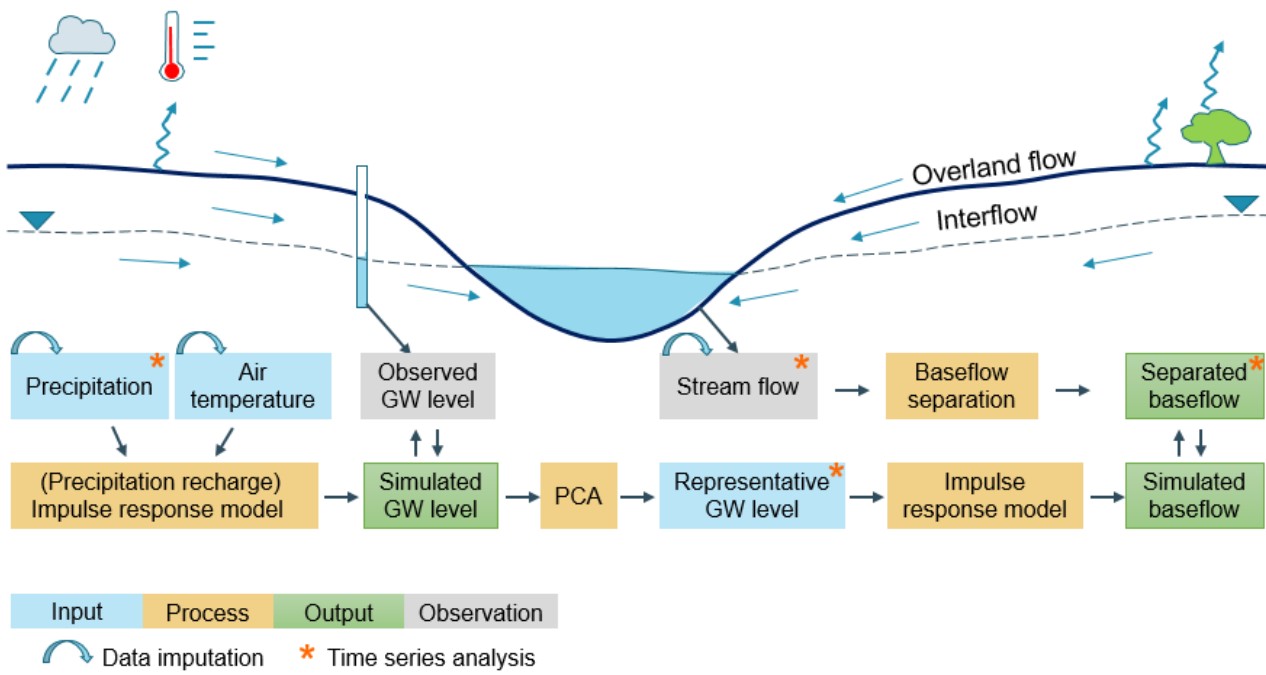

**Figure 4** An illustration of the methodology in this study. PCA is the abbreviation for principle component analysis.



An illustration of the methodology is shown in Fig. 4. As introduced in Sect. 1, we carried out a combination of impulse response modelling, baseflow separation and time series analysis to gain insights of the river–aquifer interactions in the three lowland catchments. First of all, precipitation, air temperature, groundwater level and stream flow data were collected, cleaned

and imputed where necessary. Afterwards, imputed precipitation and air temperature were used as system input for the first case use in the impulse response model, and the system output was calibrated with observed groundwater level. Only well fitted groundwater level time series were retained and further used to generate a representative groundwater level time series via principle component analysis in each catchment. This time series was applied as system input for the second case use in the impulse response model. The model output of simulated baseflow was calibrated with separated baseflow obtained from

multiple hydrograph separation approaches. At last, time series analysis was carried to investigate the trend, seasonality and the links of the relevant variables. Detailed explanations can be found in the following sections.

### 3.1 Data preparation

### 3.1.1 Data collection

Data were collected and covering one climatic cycle of 30 years (1 Jan 1990–31 Dec 2019). The climatic variables include the

daily precipitation and daily air temperature, which were obtained from KMI (Koninklijk Meteorologisch Instituut) at the closest meteorological station for each catchment (Fig. 1a). The daily stream flow time series were accessed from https://www.waterinfo.be via the wateRinfo R package interface (Van Hoey, 2020) and from the river gauging station at the catchment outlet (Fig. 1a). The groundwater levels were obtained from the piezometric observation network of INBO (Instituut voor Natuur- en Bosonderzoek), DOV (Databank Ondergrond Vlaanderen) and DEE (Département de l'Environnement et de

l'Eau). Approximately 82 % of the level observation wells are from INBO, 11 % from DOV and the rest 7 % from DEE (Fig. 5).


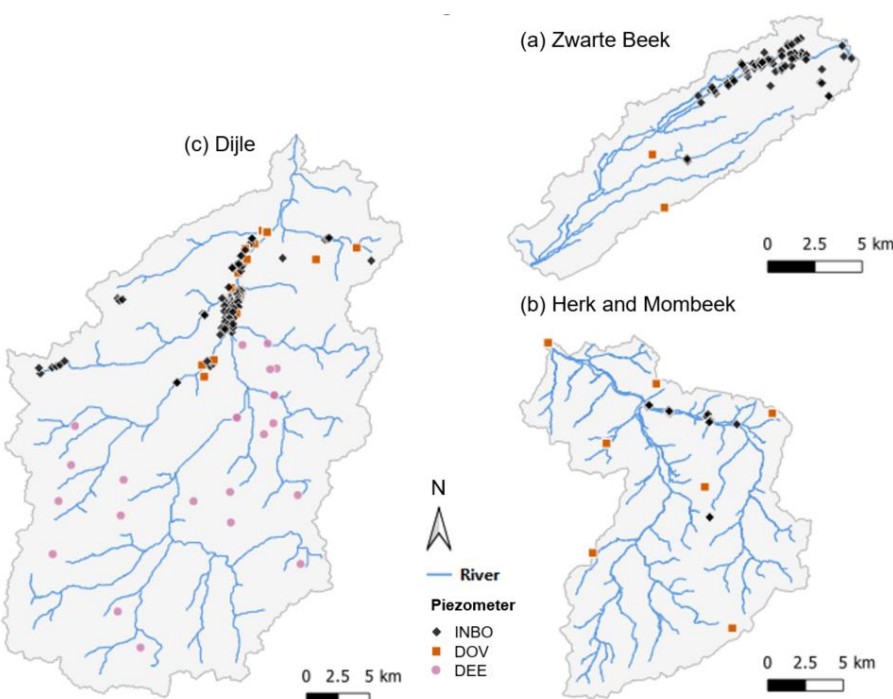

**Figure 5** Groundwater level observation wells in (a) the Zwarte Beek, (b) the Herk and Mombeek and (c) the Dijle from multiple sources (INBO, DOV, DEE).

### 3.1.2 Data cleaning

All raw time series were checked and cleaned first. For the precipitation data, the missing values were screened for further imputation. For the daily air temperatures, KMI provided daily maximum and minimum air temperatures. We took the average of the two as the daily mean air temperature. Only few scattered missing values existed in the raw data, which were filled using the value from the previous day. For the stream flow data, the gaps of the missing values were checked and few erroneous zero values were also labelled as missing values at this stage.

The raw groundwater level observations have multiple sources (INBO, DOV, DEE), time spans (from a few months to decades) and measurement intervals (e.g. daily, weekly, biweekly and monthly). From the INBO data portal, the groundwater level time series are recorded daily and the observations are from shallow aquifers, with the depth to water table less than 20 m. Mostly of the groundwater levels from the DOV data source are measured monthly, and the rest are recorded biweekly. Since we focus on river and shallow aquifer interactions, we considered a subset of the shallow aquifers by limiting the depth to water table to less than 20 m. The groundwater level observations from the DEE data portal are measured mostly on a weekly basis with the rest on a monthly basis. From the DEE data portal, we obtained very limited well observations (23) in the Walloon part of the Dijle catchment (Fig. 5c), where the elevation differences are larger than the Flanders part. Although the depths to water





table are larger than 20 m, they are still included at this point to avoid data absence in that region. Further, the length of the groundwater level observations was checked to exclude time series which covered only short periods (e.g. a few months) or very discontinuous. With the monthly frequency, time series consisted of at least 90 data points were considered to be adequate, which corresponded to one fourth of the full time period under investigation here.

### 3.1.3 Data imputation

After cleaning the available raw time series, the missing values were imputed, as required for further analysis. The imputation techniques applied in this study were (1) linear or local polynomial regression models for estimations based on different time series and (2) ordered quantile normalization for re-scaling different time series (Fan and Gijbels, 2018; Peterson and Cavanaugh, 2019).

Missing values in stream flow and precipitation are filled using both approaches. A time series of a nearby station that survived the quality checks and has less missing values, is used as the reference series to check first if a linear regression model would be adequate ($R^2 \geq 0.7$). If not satisfactory, a local polynomial regression model is applied instead. When the linear regression model is not adequate, another option is to perform the ordered quantile normalization. The marginal distribution of the target series is in this case approximated, using days with observations in both series, by transforming both with ordered quantile

normalization and making the two distributions identical in statistical properties (Peterson and Peterson, 2020).

An example below shows that when filling missing values of the stream flow in the Zwarte Beek, a linear regression model was applied first with reference stream flow data in the Mangelbeek station (Fig. 6a) and a loess regression model was used further to fill the remaining gaps with reference stream flow from the Tessenderlo station (Fig. 6b). The imputation of the

precipitation time series in the Herk and Mombeek was done using the ordered quantile normalization. Since the linear fit with the reference precipitation from the Zwarte Beek catchment was not ideal for use (Fig. 6c), matching the marginal distributions makes more sense in this case.





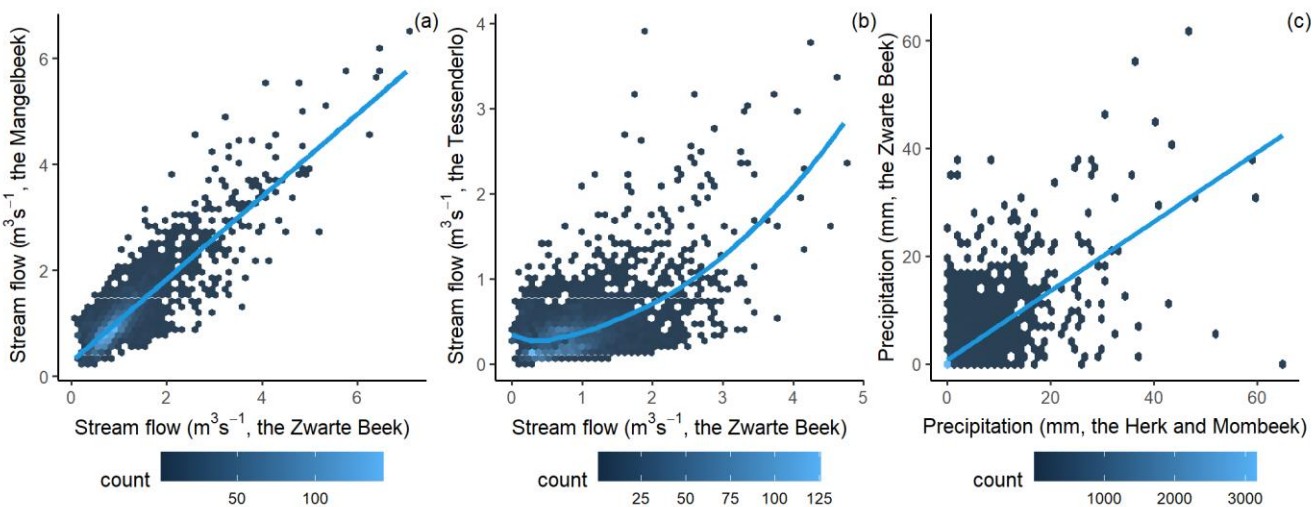

**Figure 6** Examples of different imputation approaches for the stream flow and precipitation time series: (a) stream flow in the
Zwarte Beek and the reference Mangelbeek station, $R^2 = 0.73$ and residual standard error = 0.36 m$^3$ s$^{-1}$; (b) stream flow in the
Zwarte Beek and the reference Tessenderlo station, residual standard error = 0.50 m$^3$ s$^{-1}$; (c) precipitation in the Herk and
Mombeek and the reference time series in the Zwarte Beek, $R^2 = 0.43$ and residual standard error = 3.39 mm.

### 3.2 Baseflow separation

Baseflow separation divides the stream flow into the components that originate (1) from quick flow, the sum of the interflow,
overland flow and direct precipitation in the stream network, and (2) baseflow, from delayed subsurface groundwater discharge
(Hall, 1968). The main separation techniques include (1) the graphic methods, which pick out the low-flow points from the
hydrographs and link them together as the baseflow component (Sloto and Crouse, 1996; Rutledge, 1998), and (2) the digital
filter methods, which, by applying one, two or more filters, filter out a low frequency signal representing baseflow and a high
frequency signal attributed to the quick flow component (Nathan and McMahon, 1990; Arnold and Allen, 1999; Eckhardt,
2005, 2008; Jakeman and Hornberger, 1993). The base flow index (BFI) gives the ratio of the baseflow to the total stream
flow. It is an indicator of the catchment flow regime, with high indices (>0.9) for permeable catchments with a very stable
flow regime and low indices (0.15–0.2) for impermeable catchments with a flashy flow regime (Tallaksen and Van Lanen,
2004). BFI is highly dependent on other catchment properties, such as soil type, catchment geology, hydrological and
geomorphological conditions (Tallaksen and Van Lanen, 2004).


Limitations for these hydrograph separation methods, are their intrinsic difficulty to validate the separated baseflow and the
lack of any representation of the physical processes of the river–aquifer exchange (Nathan and McMahon, 1990; Sloto and
Crouse, 1996; Batelaan and De Smedt, 2007; Killian et al., 2019). Nevertheless, it is still a fast, efficient, and widely used
approach to quantitatively estimate the groundwater discharge to river and its temporal dynamics at the catchment scale.





In this study, we used different baseflow separation methods to provide an idea on the range of potential outcomes related to hydrograph separation techniques. The following approaches were selected:

1. a series of graphical separation techniques from HYSEP (Sloto and Crouse, 1996), including fixed interval, sliding interval and local minimum technique;

2. a one-parameter digital filter from Nathan and McMahon (1990), where the filter parameter is taken as 0.925 in this

study; and

3. two-parameter filters from Eckhardt (2005, 2008), since this method agrees well with tracer based (e.g. dissolved silica) hydrograph separation in lowland catchments (Gonzales et al., 2009). The filter parameter is set as 0.98 and the $BFI_{max}$ parameter is chosen as 0.80 here, recommended by other research conducted with the same method in Flanders (Zomlot et al., 2015).


For each of the methods applied here, we consider the parameters to be constant over the entire studied period, under the assumption that the hydrological system has not evolved drastically within this time frame. This means that fitted models are mainly indicative of the meteorological forcing of the hydrological system, and any model residuals may contain effects of human interferences.

**3.3 Impulse response modelling**

**3.3.1 Model overview**

A lumped parameter impulse response model can effectively transform a hydrological system input to yield an accurate prediction of the corresponding system output, and the impulse response function (IRF) estimated in the model can provide mechanistic insights into the hydrological system, such as the peak response time and magnitude (Olsthoorn, 2007; Asmuth

and Knotters, 2004; Young, 2013). The impulse response model used in this study is the Rainfall-Response Aquifer and Watershed Flow Model (RRAWFLOW; Long,2015). This model is capable of simulating the point-measured groundwater level, stream flow, spring flow, or solute transport as system outputs, in response to a system input of precipitation, recharge, or solute injection (Long, 2015). We use it here to explore and gain insights into (1) the hydrological system response of groundwater level to meteorological forcing and (2) baseflow response to groundwater level as system input, under natural

conditions. Therefore, anthropogenic influences are not considered in the impulse response modelling here.

RRAWFLOW includes two processes, (1) the process of recharge generation from precipitation, denoted as precipitation recharge in this study, and (2) the process of precipitation recharge transitioning into a system response such as groundwater level or spring flow (Long and Mahler, 2013; Long, 2015). During the first nonlinear process, the precipitation recharge is

estimated by using a unitless soil–moisture index $s$ (Long and Mahler, 2013). This $s$ ($\leq 1.0$) represents the fraction of





precipitation that infiltrates and becomes precipitation recharge. Since the preceding rainfall events have impacts on soil moisture, the past rainfall record is counted and weighted by an exponential decay function (Jakeman and Hornberger, 1993):

$$s_i = cr_i + (1 - k_i^{-1})s_{i-1}$$

$$= c[r_i + (1 - k_i^{-1})r_{i-1} + (1 - k_i^{-1})^2 r_{i-2} + \ldots] \tag{1}$$


$$k_i = \alpha exp[(20 - T_i)f] \quad f > 0 \tag{2}$$

$$u_i = r_i s_i \tag{3}$$

$$i = 0,1,\ldots,N \quad 0 \le s \le 1$$

where $c(L^{-1})$ is a scaling coefficient to constrain $s$; $r$ is total rainfall (L); $k$ (unitless) is linked with evapotranspiration and can adjust the effect of antecedent rainfall; $i$ is the time step index; $\alpha$ (unitless) is a scaling coefficient to constrain $k$; $T$ is

mean air temperature at the land surface (° C); $f$ is a temperature modulation factor (° $C^{-1}$); and $u$ is precipitation recharge (L) (Long and Mahler, 2013).

During the second process, the response of the hydrological system to the precipitation recharge is estimated by convolution (Long, 2015). Convolution is the superposition of a series of IRFs that are initiated at the time of each impulse and scaled

proportionally by the magnitude of the corresponding impulse (Olsthoorn, 2007; Long and Mahler, 2013; Long, 2015; Asmuth et al., 2002). The discrete form of the convolution integral for uniform time steps used in RRAWFLOW is:

$$y_i = \Delta t \sum_{j=0}^{i} \beta_j h_{i-j} u_j + \psi_i + d_0 \tag{4}$$

$$i,j = 0,1,\ldots,N$$

where $h_{i-j}$ is the IRF; $u_j$ is the input; $j$ and $i$ are time step indices corresponding to system input and output, respectively; $\Delta t$

(T) is the time step duration; $N$ (unitless) is the number of time steps in the output record; $\beta_j$ (unitless) is an optional scaling coefficient of IRF; $\psi_i$ is the error component resulting from inaccuracy in measurement, sampling intervals, or model simplification assumptions; and $d_0$ is a hydraulic-head datum (L) for groundwater level simulation (Long, 2015). In this process, precipitation recharge ($u_j$) is assumed to be the only forcing that can cause an increase in the hydraulic head to be above $d_0$ (Long, 2015).


Hydrological system dynamics can be approached by different types of parametric IRFs. In this study, we use parametric gamma functions, as they tend to work well and allow us to easily evaluate the fitted parameters. The gamma function and gamma distribution functions in RRAWFLOW are:

$$\Gamma(\eta) = \sum_{t=0}^{\infty} t^{\eta-1} e^{-t} dt \tag{5}$$


$$\gamma(t) = \frac{\lambda^\eta t^{\eta-1} e^{-\eta t}}{\Gamma(\eta)} \tag{6}$$

$$h(t) = \epsilon \gamma(t) \tag{7}$$

where $\Gamma$ and $\gamma$ are gamma function and gamma distribution function, respectively; $\lambda$ (unitless) and $\eta$ (unitless) are shape parameters; $t$ (T) is the time centred on each discrete time step; $h$ is the scaled gamma distribution function; and $\epsilon$ (unitless)



is the scaling coefficient that compensates for the system response when there is not a one–to–one relation between system input and output (Olsthoorn, 2007; Long, 2015).

RRAWFLOW allows the use of two superposed gamma distribution functions, which represent the components of quick flow and slow flow in the hydrological system, and it also allows the system records to be divided into two periods, namely dry and wet periods (Long, 2015). Thus time–variant double–gamma IRFs can be defined, if necessary, to simulate the dominant

transient characteristics of the system.

The "L-BFGS-B" method is used for RRAWFLOW optimization, which allows working with lower and upper parameter bounds (Byrd et al., 1995). The wrapper around RRAWFLOW for executing the optimization was further paralleled across the different time series, to enable processing larger sets of time series at once. The Nash–Sutcliffe Efficiency (NSE) is used

for the evaluation of the simulated time series (Nash and Sutcliffe, 1970). NSE = 1 means a perfect model fit, and NSE = 0 indicates that the model has the same predictive power as the mean of the time series in terms of the sum of the squared error, and if NSE < 0, it is worse than the observed mean (Nash and Sutcliffe, 1970).

The model parametrization and modifications to the original RRAWFLOW code are discussed for the two use cases in the

following Sect. 3.3.2 and 3.3.3.

### 3.3.2 Groundwater level response modelling

The groundwater level response modelling consists of two processes, (1) precipitation recharge generation and (2) recharge transitioning into groundwater level. The system input is precipitation and air temperature data, and the system output is compared with groundwater level observations. The model time–step interval is daily, which is determined by the input time

series of precipitation and air temperature. A warm-up period of six years was added (1984–1989), using the data record from the first six years (1990–1995). This estimated warm–up period can largely reduce the antecedent effects of the system before it is fully incorporated into the simulation.

To better represent the dominant transient characteristics of the hydrological system, the dry (April–September) and wet

(October–March) periods were defined, mainly because of very different evapotranspiration effects within the year (Fig. 2). Different gamma distribution functions are used for representing quick and slow components as well. In total, four gamma distribution functions are used in the simulation, representing the quick flow and slow flow processes during both the dry and wet periods. In each gamma distribution function, there are three parameters to be optimized, two shape parameters ($\lambda$ and $\eta$, Eq. 6) and one scaling coefficient ($\epsilon$, Eq. 7). The double–gamma IRFs representing dry ($h_{dry}$) and wet ($h_{wet}$) periods are

shown below:





$$h_{dry}(t) = \epsilon_1 \frac{\lambda_1^{\eta_1} t^{\eta_1-1} e^{-\eta_1 t}}{\Gamma(\eta_1)} + \epsilon_2 \frac{\lambda_2^{\eta_2} t^{\eta_2-1} e^{-\eta_2 t}}{\Gamma(\eta_2)} \qquad (8)$$

$$h_{wet}(t) = \epsilon_3 \frac{\lambda_3^{\eta_3} t^{\eta_3-1} e^{-\eta_3 t}}{\Gamma(\eta_3)} + \epsilon_4 \frac{\lambda_4^{\eta_4} t^{\eta_4-1} e^{-\eta_4 t}}{\Gamma(\eta_4)} \qquad (9)$$

where the subscripted number 1 and 2 for quick and slow components during dry period, and 3 and 4 for quick and slow components during the wet period. Besides the 12 parameters from the IRFs (Eq. 8–9), the hydraulic-head datum parameter ($d_0$, Eq. 4) is also included for optimization.

Although the IRFs can have infinite length, we define the system has a maximum memory of six years for an impulse, to avoid long–term trends or human interference effects to bias the fitted IRFs. The whole 30–year period is used for model calibration, as we are doing exploratory rather than confirmatory analysis here.

### 3.3.3 Baseflow response modelling

The baseflow response modelling simulates the process of groundwater discharge to stream flow, with the groundwater level as system impulse and the baseflow as system response. Fitted groundwater levels from the previous step are used for this process. However, as there are many groundwater level time series in each catchment, we introduce a single representative groundwater level time series, to reduce the dimensionality of the problem. The representative groundwater level time series was obtained by extracting the first principle component of the adequately fitted groundwater level time series from the previous step, to represent the catchment status in terms of groundwater level.

The model was set up with the representative groundwater level time series as the input data and separated baseflow from the different separation techniques was used as the system response observations. A six–year warm–up period (1984-1989) was again added, using the representative groundwater level time series for the period between 1990 and 1995. As the groundwater discharge to stream flow happens beneath the land surface, without the pronounced wet–dry period distinctions from the effects of evapotranspiration, a time–invariant model is assumed for this modelling process. Only two gamma distribution functions are thus used, for representing the quick and slow components. The compound IRF ($h_{comp}$) is shown below:

$$h_{comp}(t) = \epsilon_q \frac{\lambda_q^{\eta_q} t^{\eta_q-1} e^{-\eta_q t}}{\Gamma(\eta_q)} + \epsilon_s \frac{\lambda_s^{\eta_s} t^{\eta_s-1} e^{-\eta_s t}}{\Gamma(\eta_s)} \qquad (10)$$

where the subscripted letter q and s for quick and slow components, respectively.

Furthermore, a modification is made to the original code, where a constant drainage level is subtracted from the input groundwater level. This operation turns the representative groundwater level into a proxy for the hydraulic gradient. The modified convolution integral has the form as below:

$$y_i^{mod} = \Delta t \sum_{j=0}^{i} \beta_j h_{i-j}(u_j - d_0) + \psi_i \qquad (11)$$



There are hence seven parameters to be optimized: three parameters ($\lambda$, $\eta$ and $\epsilon$) for each of the two gamma distribution functions and the drainage level, which was actually implemented by using the hydraulic-head datum $d_0$. This allowed the modification to be made with minimal code adjustments.

### 3.4 Time series analysis

Time series analysis is used as a tool to extract meaningful statistics and other characteristics of the collected data. In this study, we apply (1) time series decomposition to study the temporal evolution, trend and seasonality of the variables over 30–year period and (2) cross correlation analysis to explore the potential links between them.

The studied hydrological variables include precipitation, representative groundwater level, stream flow and baseflow. Besides
these, we added some characteristic values derived from stream flow and representative groundwater level time series to capture their evolution over time as well. The exceedance percentiles of 10 %, 50 %, and 90 % for the stream flow and representative groundwater level time series are calculated over a sliding window of 365 days, representing low–flow and low–level, median–flow and median–level, high–flow and high–level time series, respectively. For the groundwater level, this is similar to the characteristic levels typically assessed in the framework of ecohydrological studies, albeit with different
percentiles and window sizes (Lammerts et al., 2001).

### 3.4.1 Time series decomposition

Hydrological time series usually have strong seasonal variations and can be decomposed using an additive approach to study the temporal evolution of the different components. In this study, we apply the Seasonal and Trend decomposition with Loess (STL) to decompose the time series into three components: (1) a trend-cycle component, (2) a seasonal component, and (3) a
reminder component (Cleveland et al., 1990). The additive decomposition of the target time series has the following form:

$$y_t = T_t + S_t + R_t \tag{12}$$

where $y_t$ is the seasonal time series; $T_t$ is the smoothed trend-cycle component; $S_t$ is the seasonal component; and $R_t$ is the remainder component (Cleveland et al., 1990).

The features of the STL decomposition can be summarized using two parameters:

    1. the strength of the trend:

$$F_t = max(0, 1 - \frac{Var(R_t)}{Var(T_t + R_t)}) \tag{13}$$

with $F_t$ close to zero, representing a weak trend, and closer to one, representing a strong trend, and

    2. the strength of the seasonality:


$$F_s = max(0, 1 - \frac{Var(R_t)}{Var(S_t + R_t)}) \tag{14}$$

with $F_s$ close to zero indicates a weak seasonality, and close to one indicates a strong seasonality.





### 3.4.2 Cross correlation analysis

As there are intrinsic associations within the studied hydrological variables, we apply cross correlation analysis to measure their linear correlation between two relevant variables, which can help building links with or explaining other performed analysis in this study. The applied formula for Pearson's correlation coefficient ($r_{xy}$) is (Pearson, 1896):

$$r_{xy} = \frac{\sum_{i=1}^{n}(x_i-\overline{x})(y_i-\overline{y})}{\sqrt{\sum_{i=1}^{n}(x_i-\overline{x})^2}\sqrt{\sum_{i=1}^{n}(y_i-\overline{y})^2}} \tag{15}$$

where $x_i$ and $y_i$ are the individual sample points indexed with i, $\overline{x}$ and $\overline{y}$ are the sample means of x and y variables, and n is the sample size.

## 4 Results

### 4.1 Separated baseflow

Figures 7 to 9 show the temporal evolution of the separated baseflow on a daily basis and the comparison of the stream flow and baseflow by different methods in the three catchments. The fixed interval and sliding interval approaches yield slightly higher estimated mean BFIs than other methods, since they capture quite a large amount of baseflow from the peak flows and high flow periods in the stream flow record (Fig. 7a–b; Fig. 8a–b; Fig. 9a–b). The local minimum and Eckhardt methods result in slightly lower mean BFIs than the previous two methods, as they tend to filter out large impact of high stream flow events on the baseflow (Fig. 7c–d; Fig. 8c–d; Fig. 9c–d). The estimated mean BFI from the Nathan approach is the lowest among all the methods, and the separated baseflow time series has less amplitude variation over time and is less influenced by high stream flows than the other methods (Fig. 7e; Fig. 8e; Fig. 9e). The mean BFIs over the 30–year period range between 0.73–0.86 for the Zwarte Beek, 0.70–0.82 for the Herk and Mombeek and 0.77–0.84 for the Dijle.



**Figure 7** The separated baseflow and stream flow (in grey) time series over the 30–year study period in the Zwarte Beek.







**Figure 8** The separated baseflow and stream flow (in grey) time series over the 30–year study period in the Herk and Mombeek.

**Figure 9** The separated baseflow and stream flow (in grey) time series over the 30–year study period in the Dijle.

## 4.2 Impulse response modelling of the hydrological system

### 4.2.1 Groundwater level response modelling

Altogether, 100 groundwater level time series in the Zwarte Beek, 45 in the Herk and Mombeek and 198 in the Dijle were

used for the groundwater level response modelling. The distribution of the calculated NSEs is shown in Fig. 10. The positive

NSEs (0–1) of the simulated time series take up 89 %, 95 % and 83 % of all the simulations in the Zwarte Beek, the Herk and

Mombeek and the Dijle, respectively. The overall model performance is better in the Zwarte Beek and the Herk and Mombeek

than the Dijle, as can be seen in the median NSE values (Fig. 10). Taking into account the balance between evaluating model

performance and allowing sufficient simulated time series to compute the representative groundwater level, we considered the





simulations with a NSE > 0.3 as satisfactory. This allowed 70 simulated time series for the Zwarte Beek, 38 for the Herk and
Mombeek and 108 for the Dijle to be retained. Examples of the retained groundwater level time series for each catchment are
shown in Fig. 11. The time series for the Zwarte Beek and the Herk and Mombeek (Fig. 11a-b) seem to be reproduced by the
impulse response model in a very detailed way (NSEs > 0.8), while the time series for the Dijle (Fig. 11c) seems to exhibit
very low levels at the end of summer or begin of autumn, and apparently is not straightforward to be captured with the impulse
response modelling.

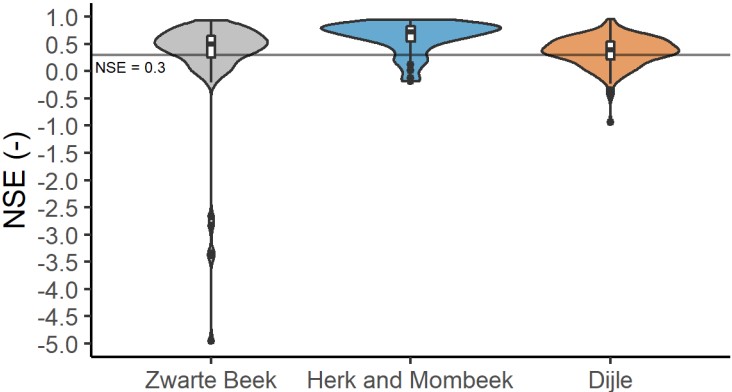

**Figure 10** NSEs of the simulated groundwater level time series in the three catchments.

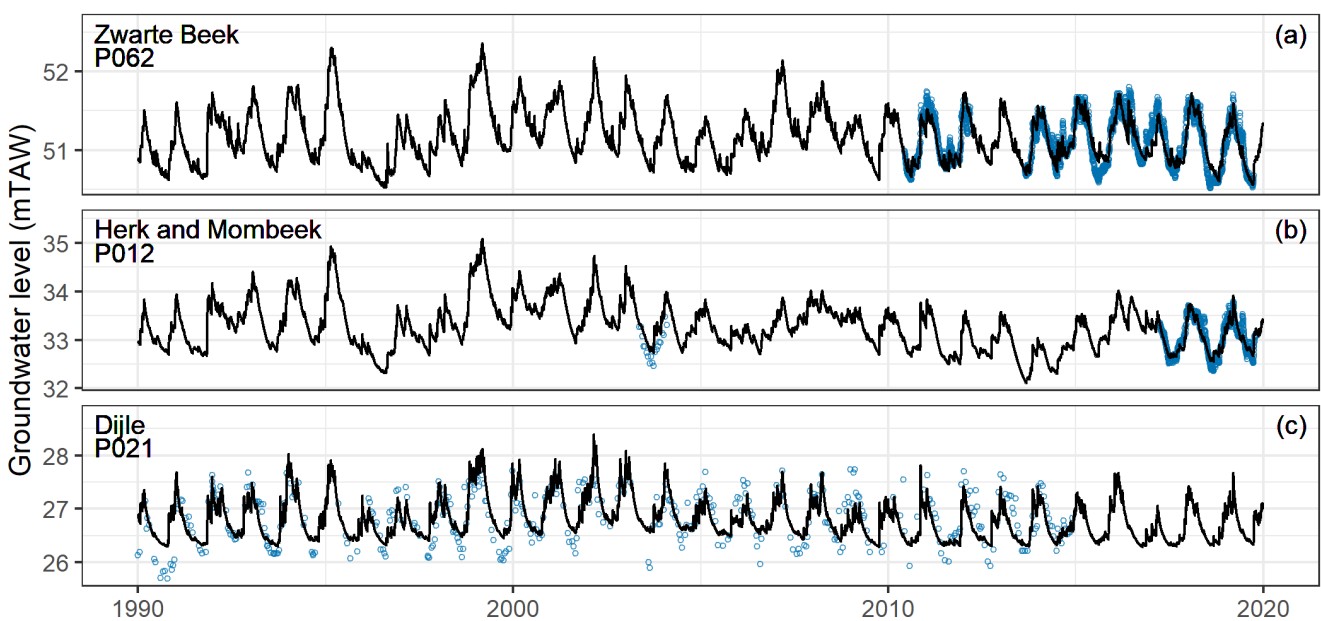




**Figure 11** Selected observed (in blue) and simulated (in black) groundwater level time series for impulse response modelling: (a) ZWAP062 in the Zwarte Beek (NSE = 0.838), (b) MOMP012 in the Herk and Mombeek (NSE = 0.857) and (c) DYLP021 in the Dijle (NSE = 0.583).

Figure 12 shows two time series with low NSEs, that were not retained in the further analyses, for illustration purposes. The first one (Fig. 12a) seems to plateau in the wet season (winter), which is typical for very shallow groundwater tables, or specific drainage conditions. This can however not be reproduced by our current impulse response modelling approach. Additionally, there are some outliers that are difficult to explain with something other than human interferences. The second time series (Fig. 12b) is a clear illustration of the latter as well, where most likely changes were made to the drainage or surface water network

at different points over the 30 years. While the model residuals could be exploited further to quantify these effects, this is considered out of scope here. These changes also bias the model fits, which makes it difficult to extract the "natural" signal.

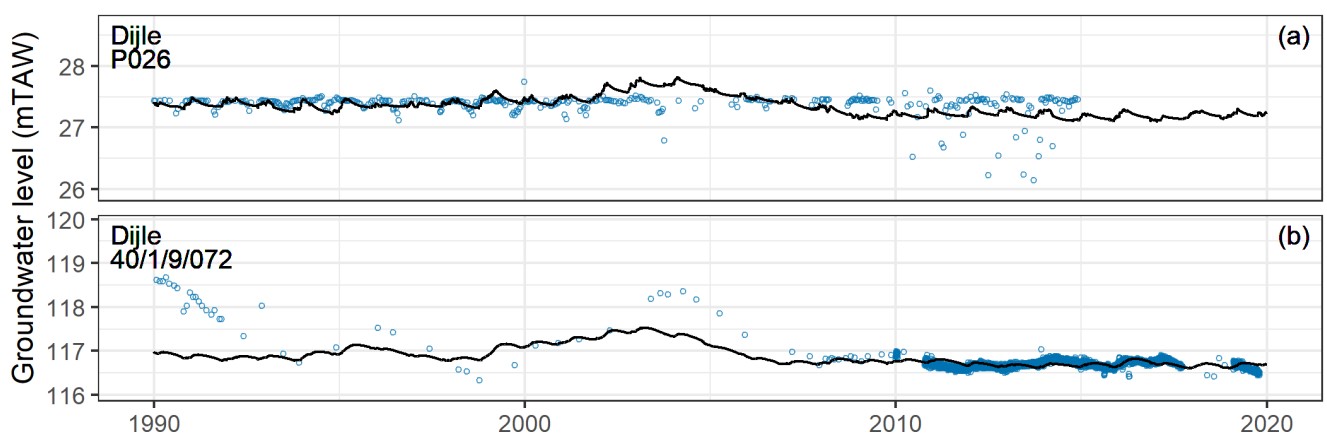

**Figure 12** Two simulated groundwater level time series with low NSEs, (a) DYLP026 (NSE = -0.404) and (b) 40/1/9/072 (NSE = 0.225) in the Dijle.


The IRF curves from the modelling are shown in Fig. 13 for each retained groundwater level time series in the three catchments. Based on these IRFs, we extracted the peak time, representing the time it takes for the groundwater level response to reach its maximum value due to precipitation recharge, and the corresponding peak response (Fig. 14). To explore the spatial patterns and identify potential factors that influence the hydrological system response, we plotted the paired variables and added linear

trend lines as well (Fig. 15).





**Figure 13** IRFs derived from the groundwater level response modelling for dry (April–September) and wet (October–March) periods in the three catchments.





**Figure 14** Peak groundwater level response derived from the groundwater level response modelling for dry (April–September) and wet (October–March) periods in the three catchments. Black triangles represent the mean values and orange triangles represent the median values.







**Figure 15** Changes in peak time and peak response for each retained groundwater level time series under the potential influences of the well distance to the river and the depth to water table. Linear regression lines with a 95 % confidence interval are added in each plot, for a quick estimation and visualization of the possible trend between the two variables.

In the Zwarte Beek, approximately 73 % (51/70) of the retained groundwater level wells in the dry period (April–September)
and 61 % (43/70) in the wet period (October–March) reach its peak response in the first day (Fig. 13a–b; Fig. 14a–b). This indicates a very fast response to precipitation recharge in the shallow aquifer. The rest of the level time series has the peak time ranging between 2 and 548 days during the dry period (Fig. 14a) and between 2 and 98 days during the wet period (Fig. 14b). On average, it takes 45 days in the dry period (Fig. 14a) and 10 days in the wet period (Fig. 14b) for the shallow aquifer to obtain its maximal response to precipitation recharge. For the response magnitude, 1 mm of precipitation recharge can cause
a maximal immediate groundwater level rise between 0.01 and 25.7 mm during the dry period (Fig. 14a), and between 0.8 and 14.0 mm during the wet period (Fig. 14b). The mean peak responses are 3.9 and 5.3 mm for the dry and wet seasons, respectively (Fig. 14a–b). Therefore, relatively higher immediate increase of groundwater level occurs in the wet than the dry period. During the dry period, we did not observe clear impacts of the well distance to the river and the depth to water table on the peak time and response (Fig. 15a–b, 15e–f). During the wet period, there seems to be a slight increase in peak time
when the well is farther from the river and the depth to the water table is larger (Fig. 15c–d), while no significant impacts of these two factors on the peak response are observed (Fig. 15g–h).

In the Herk and Mombeek, around 42 % (16/38) of the groundwater level time series in the dry period and 71 % (27/38) in the wet period reach their peak response in the first day (Fig. 13c–d; Fig. 14c–d). For the rest, the range of the peak time is larger
in the dry period (2–570 days; Fig. 14c) than the wet period (2–104 days; Fig. 14d). The mean peak time is 81 and 15 days for the dry and wet periods, respectively (Fig. 14–d). Therefore, there is a clear trend of much faster response of the shallow aquifer to precipitation recharge in the wet period than the dry period. Regarding the peak response magnitude, the range is between 0.8 and 18.4 mm for 1 mm of precipitation recharge in the dry period (Fig. 14c), and between 2.6 and 32.3 mm in the wet period (Fig. 14d). The mean peak magnitudes are 5.4 and 12.0 mm for dry and wet periods, respectively (Fig. 14c–d).
Moreover, if an observation well is closer to the river, the peak time is shorter and the peak response is higher (Fig. 15a, c, e, g), regardless of dry or wet periods. Similar patterns are also shown for the factor of the groundwater depth. If the depth to water table is smaller, the peak time is shorter and the peak response is relatively higher (Fig. 15b, d, f, h).

During the dry period in the Dijle catchment, approximately 68 % (73/108) of the observation wells reach their peak response
in the first day while the remaining ones vary between 2 and 574 days (Fig. 13e; Fig. 14e). During the wet period, around 59 % (64/108) reach their peak response in the first day, and the rest peaks between 2 and 528 days (Fig. 13f; Fig. 14f). The mean peak time is 159 and 27 days for the dry and wet periods, respectively (Fig. 14e–f). The peak response ranges between 0.02 and 31.1 mm for the dry period (Fig. 14e), and between 0.6 and 36.1 mm for the wet period (Fig. 14f) for 1 mm of precipitation



recharge. The mean peak response magnitudes are 4.9 and 9.2 mm for the dry and wet periods, respectively (Fig. 14e–f). We
did not observe clear impacts of the distance to the river nor the depth to water table on the peak time (Fig. 15a–d). Although
there seem to be a positive trend in these subfigures (Fig. 15a-d), the linear fits seem to be heavily influenced by outliers and
the uncertainties of the fits are large. However, there seems to be a negative correlation between peak response and depth to
water table (Fig. 15d, h), where the increase of depth leads to the decrease of peak response. The link between peak response
and distance to the river is not obvious.

**4.2.2 Baseflow response modelling**

The retained groundwater level time series from the previous simulations are used to generate a time series of the groundwater
level for representing groundwater storage over the entire catchment. The extracted first principle component of the simulated
groundwater levels takes up 83 %, 91 % and 79 % of the total variance, for the Zwarte Beek, the Herk and Mombeek, and the
Dijle, respectively. The representative groundwater level time series has a mean of zero and the level is rescaled to a relative
elevation (in meter) since only the level differences are relevant for the baseflow response modelling (Fig. 16).

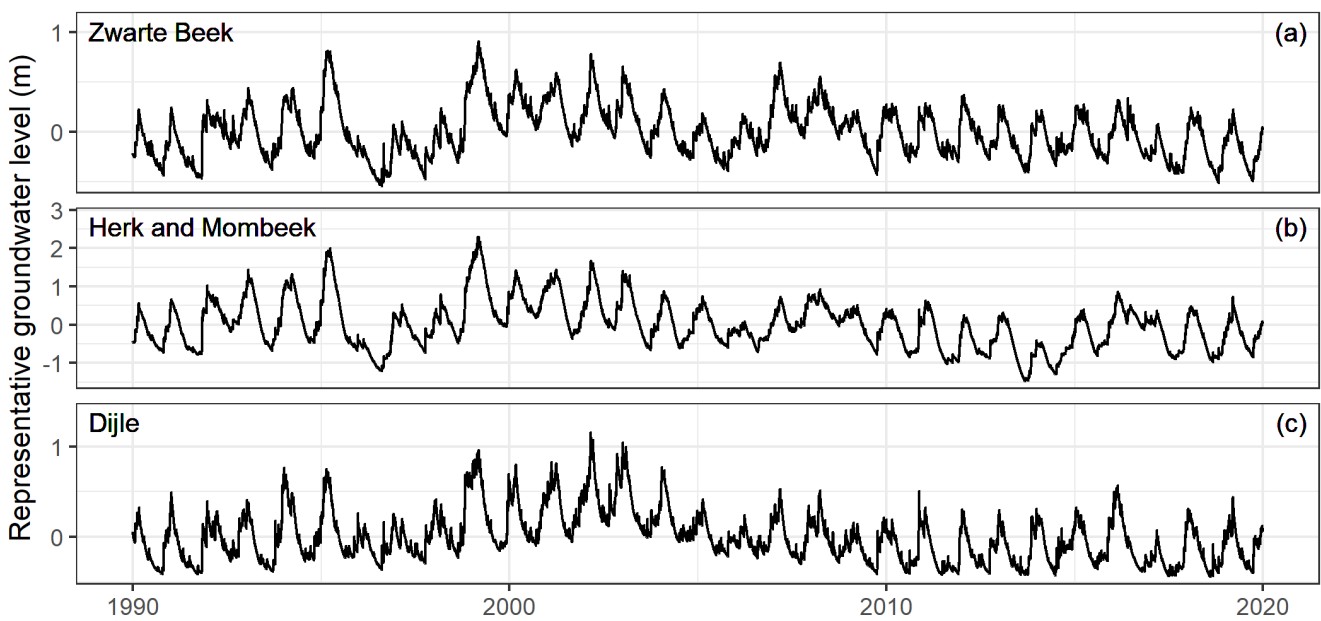

**Figure 16** The representative groundwater level (relative elevation) time series in each catchment.

The mean NSEs of simulated baseflow time series are 0.325 for the Zwarte Beek, 0.384 for the Herk and Mombeek and 0.146
for the Dijle. The Eckhardt and Nathan methods work better than other methods in the Zwarte Beek, with a NSE of 0.385 and
0.377, respectively. The Nathan method yields a higher NSE of 0.478 than other methods in the Herk and Mombeek. In Dijle,
the Eckhardt method performs better than other methods with a NSE of 0.212. The IRFs extracted from the modelling show





that there is basically an immediate response in the first time step, and the responses for a few days are already negligible. This means that our impulse response model basically falls back to a linear regression model of the baseflow in function of the representative groundwater level, where the intercept and the slope are related to the drainage level and impulse response function value for the first time step, as shown in Table 1. The mean peak response, calculated as the mean value of the slopes in Table 1, shows an average change of 0.95 m$^3$ s$^{-1}$ of baseflow in the Zwarte Beek, 0.69 m$^3$ s$^{-1}$ in the Herk and Mombeek and 2.2 m$^3$ s$^{-1}$ in the Dijle, as a response of 1 m change in the groundwater level.

**Table 1** The linear regression fit of the simulated baseflow as a function of the representative groundwater level in each catchment. Level = representative groundwater level; Fixed inverval, Sliding interval, Local minimum, Eckhardt and Nathan refer to the baseflow separated from these separation techniques.

| Catchment | Fixed interval–Level | Sliding interval–Level | Local minimum–Level | Eckhardt–Level | Nathan–level |
|---|---|---|---|---|---|
| Zwarte Beek | y = 1.03 x + 0.92 | y = 1.04 x + 0.92 | y = 0.98 x + 0.89 | y = 0.92 x + 0.79 | y = 0.8 x + 0.71 |
| Herk and Mombeek | y = 0.75 x + 1.02 | y = 0.76 x + 1.02 | y = 0.69 x + 0.97 | y = 0.74 x + 0.98 | y = 0.52 x + 0.81 |
| Dijle | y = 2.16 x + 5.3 | y = 2.26 x + 5.3 | y = 2.07 x + 5.22 | y = 2.65 x + 5.08 | y = 1.88 x + 4.83 |

The corresponding simulated baseflow does seem to capture some of the variations of the separated baseflow time series, but tends to be smoother, and the baseflow peaks are not well reproduced (Fig. 17–19).

For the Zwarte Beek, the simulated baseflow in the first two decades (1990–2009) can capture most of the low baseflow points, while for the last decade (2010–2019), there are some continuous periods with overestimation (Fig. 17). In the Herk and Mombeek catchment, the simulated time series seem to match well with the separated baseflow between 2005 and 2010 and major model overestimation occurs in the first decade (Fig. 18). For the Dijle catchment, the differences between the simulated and separated baseflow are relatively larger than the other catchments. There are continuous periods with either overestimation (1996–1999) or underestimation (2000–2008) of the baseflow (Fig. 19). The mean BFIs of the simulated time series, after excluding the over fits (where BFI > 1), range between 0.64–0.72 for the Zwarte Beek, 0.59–0.62 for the Herk and Mombeek and 0.71–0.73 for the Dijle.





**Figure 17** The comparison between the separated baseflow and simulated baseflow (in black) in the Zwarte Beek.





**Figure 18** The comparison between the separated baseflow and simulated baseflow (in black) in the Herk and Mombeek.

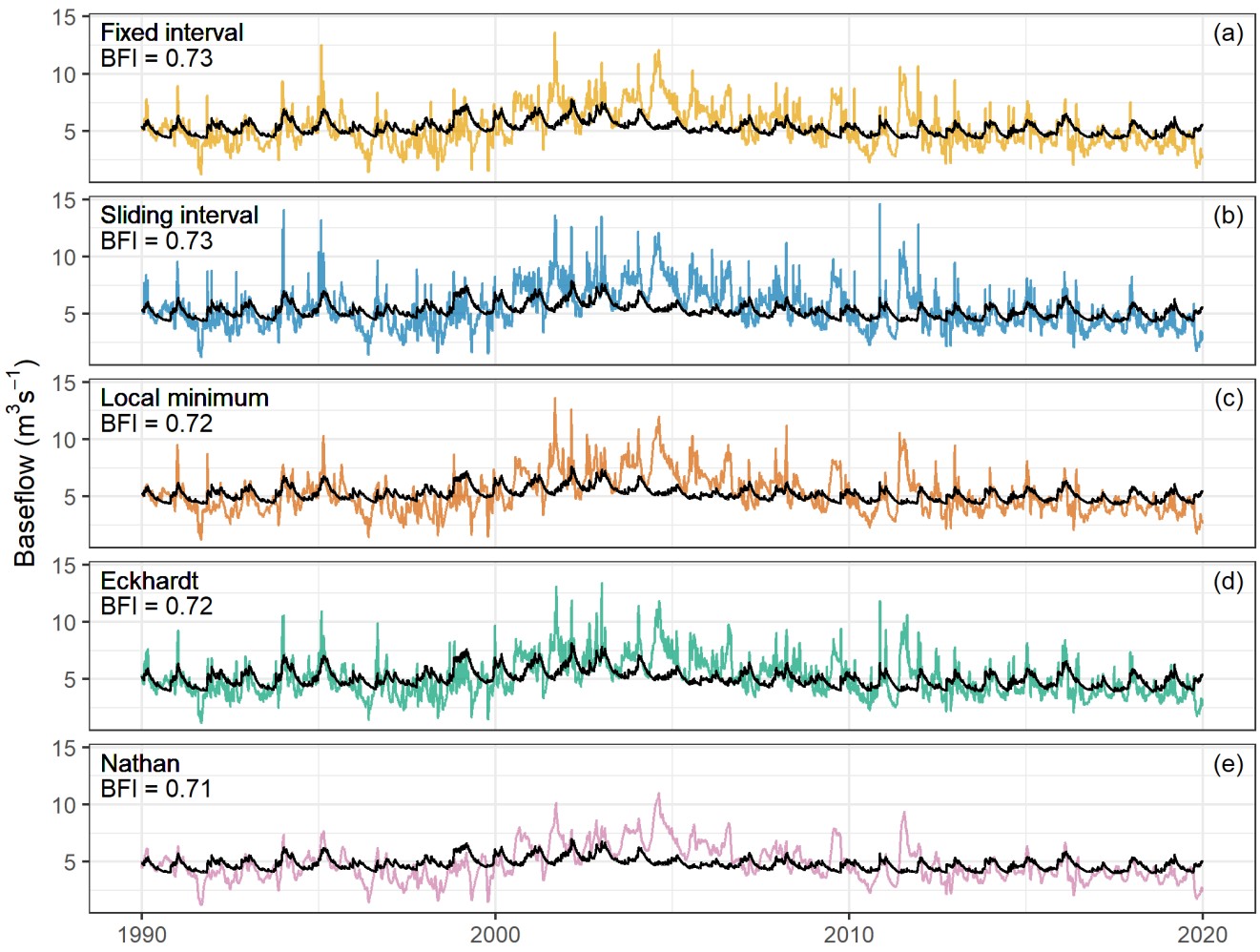

**Figure 19** The comparison between the separated baseflow and simulated baseflow (in black) in the Dijle.

**4.3 Time series analysis**

**4.3.1 Time series decomposition**

The trend analysis uses a window of 365 consecutive daily observations for estimating the trend–cycle component on a yearly basis. This span allows the retention of sufficient fluctuation in the data. The seasonal component analysis takes 365 days as its periodic cycle and a window of 11 consecutive years for estimating the seasonality.

Precipitation has little trend over the 30–year period and the magnitudes of the extracted trend components are much smaller than the raw time series (Fig. 20a; Fig. 21a; Fig. 22a; Fig. 23). There is some level of seasonality observed for the precipitation,





however, the amplitudes of the seasonal components vary a lot between different years (Fig. 20b; Fig. 21b; Fig. 22b; Fig. 23). Therefore, the seasonality of precipitation is relatively weak.

The representative (Rep.) groundwater level time series has strong trend and seasonal strengths, and locates in the cluster formed by the individual (Indiv.) groundwater level time series, which indicates that it is a good representation of the catchment status in terms of groundwater level (Fig. 23). The trend component curve itself does not show a continuous positive nor negative trend over the whole 30–year period, but rather demonstrates varying trend directions at different periods (Fig. 20c; Fig. 21c; Fig.22c). The seasonal strength of the representative groundwater level is close to that of the air temperature (Fig.
23).The characteristic levels are also strong in trend and the variations of the trend components are similar to the representative groundwater level (Fig. 20c; Fig. 21c; Fig. 22c). However, the seasonalities are largely reduced if compared with the representative groundwater level, due to the fact that they were extracted from the representative groundwater level using only certain percentiles (10 %, 50 %, and 90 %) and smoothed after a moving sliding window of 365 days.

The stream flow demonstrates some level of trend and seasonality (Fig. 20e–f; Fig. 21e–f; Fig. 22e–f). For the characteristic flows, there is a decrease in seasonality and an increase in trend when compared to the stream flow features (Fig. 23). For instance, the high–flow, which takes the exceedance percentile of 90% based on the raw stream flow records, has its trend strength enlarged and its seasonal strength reduced (Fig. 23).

The trend and seasonal strengths of the separated baseflow time series are medium, and the different baseflow time series fall somewhere between the stream flow and representative groundwater level in these plots (Fig. 20g–h; Fig. 21g–h; Fig. 22g–h; Fig. 23). This indicates that there is a range of outcomes, in terms of trend and seasonal components, that varies from more similar to the stream flow time series characteristics to closer to the representative groundwater level characteristics. Under the assumption of a constant drainage level, it seems that the Nathan method, which provides results closer to the representative
groundwater level, is more appropriate in this setting.





**Figure 20** The trend and seasonal components of the decomposed precipitation, representative groundwater level, stream flow, baseflow, and the characteristic values of groundwater level and stream flow in the Zwarte Beek.





**Figure 21** The trend and seasonal components of the decomposed precipitation, representative groundwater level, stream flow, baseflow, and the characteristic values of groundwater level and stream flow in the Herk and Mombeek.

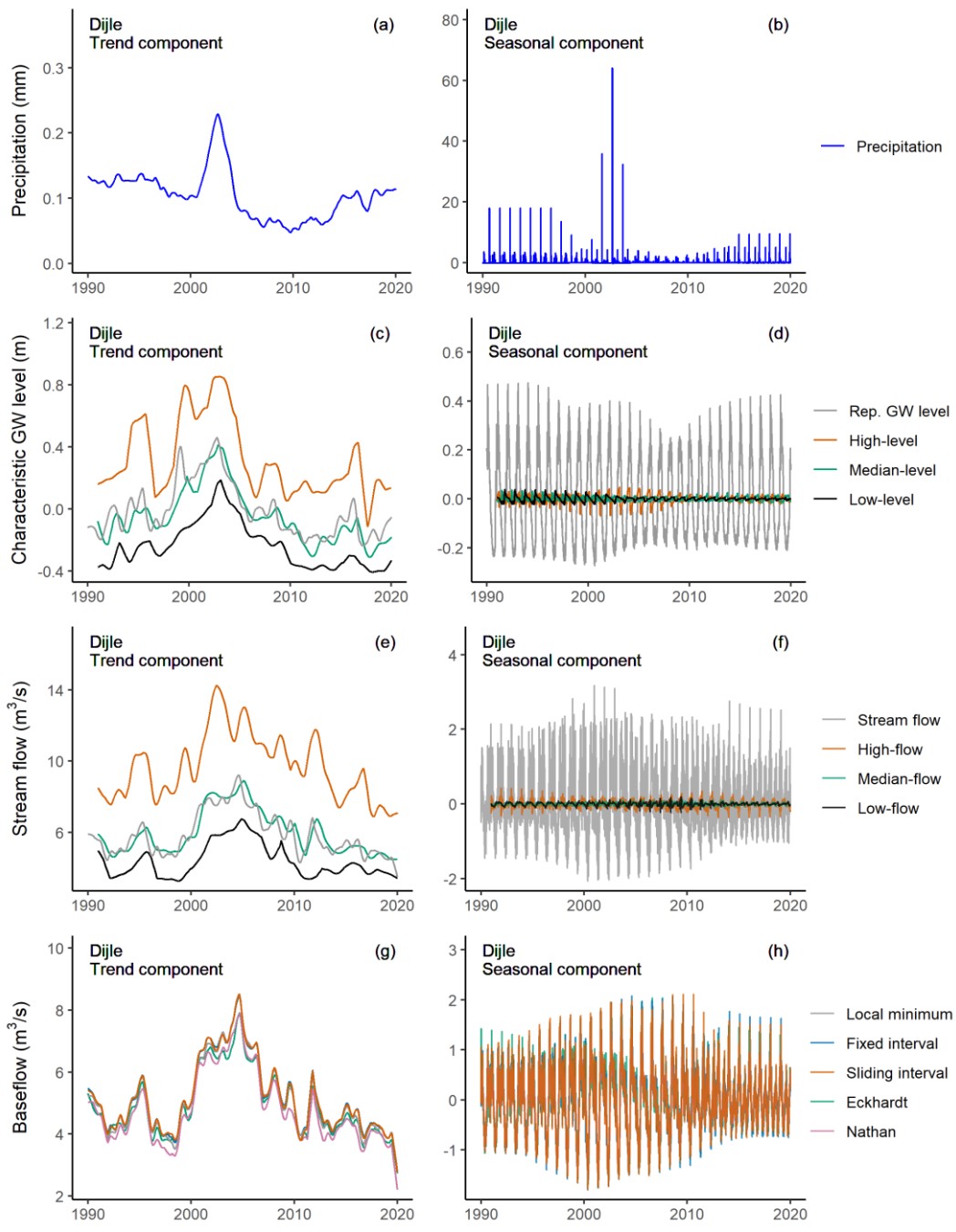

**Figure 22** The trend and seasonal components of the decomposed precipitation, representative groundwater level, stream flow, baseflow, and the characteristic values of groundwater level and stream flow in the Dijle.





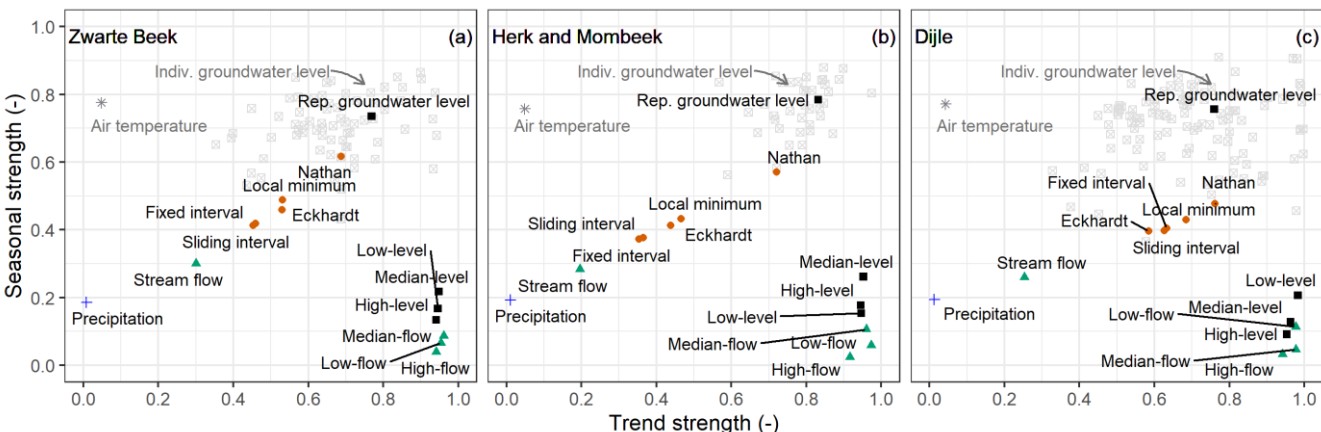


**Figure 23** Trend and seasonal strengths of the precipitation, air temperature, representative groundwater level, individual groundwater level, separated baseflow, and the characteristic values of groundwater level and stream flow in the three catchments.

### 4.3.2 Cross correlation analysis

The calculated Pearson's correlation coefficients between the paired variables are listed in Table 2. The precipitation and representative groundwater level do not seem not be linearly correlated (max. 0.13). On the contrary, the precipitation and stream flow are relatively well correlated (0.48-0.63). The groundwater level and stream flow are more correlated in the Zwarte Beek and Herk and Mombeek than in the Dijle catchment. The correlation coefficients between the representative groundwater level and the separated baseflow are relatively higher in the Zwarte Beek and Herk and Mombeek, while the values are

relatively lower in the Dijle. Among all the separation methods, the correlation between the groundwater level and baseflow from the Eckhardt and Nathan methods seems to be stronger than from other methods in the Zwarte Beek and Herk and Mombeek, while in Dijle, the baseflow from the Eckhardt method seems to be more correlated with the groundwater level than other methods.

**Table 2** Cross correlation coefficients between paried variables in the three catchments. PPT = precipitation; Level = representative groundwater level; Local minimum, Fixed inverval, Sliding interval, Eckhardt and Nathan refer to the baseflow separated from these separation techniques.



| Catchment | PPT–Level | PPT–stream flow | Level–stream flow | Level–Fixed interval | Level–Sliding interval | Level–Local minimum | Level–Eckhardt | Level–Nathan |
|---|---|---|---|---|---|---|---|---|
| Zwarte Beek | 0.10 | 0.48 | 0.48 | 0.55 | 0.55 | 0.57 | 0.62 | 0.61 |
| Herk and Mombeek | 0.07 | 0.46 | 0.45 | 0.57 | 0.58 | 0.61 | 0.64 | 0.69 |
| Dijle | 0.13 | 0.63 | 0.36 | 0.36 | 0.37 | 0.35 | 0.46 | 0.35 |

## 5 Discussion

### 5.1 Baseflow estimation

As an important indicator of river–aquifer interactions, baseflow is estimated from two different perspectives in this study, one from the stream flow and the other from the groundwater flow, in terms of groundwater level. The latter approach does still rely on the first however, as separated baseflow is used as the dependent variable in the modelling exercise. These two angles have their own limitations and strengths for estimating the baseflow.

Baseflow separation from a stream flow hydrograph is a subjective process which is based on different mathematical techniques rather than the physical processes governing exchange between river and aquifer (Nathan and McMahon, 1990; Sloto and Crouse, 1996; Batelaan and De Smedt, 2007; Killian et al., 2019). Three methods from HYSEP (fixed interval, sliding interval and local minimum) select the minimum values of the hydrograph with an interval by different algorithms (Sloto and Crouse, 1996). Alternatively, the Nathan and Eckhardt methods use parameters which have some physical connection with the catchment characteristics, such as flow status of streams within the year (perennial or ephemeral) and the permeability of the aquifers (Nathan and McMahon, 1990; Arnold and Allen, 1999; Eckhardt, 2005).

Despite the methodological differences, the mean BFIs of the separated baseflow time series are never less than 0.70 in the three catchments (Fig. 7–9). These high BFIs indicate that the studied catchments are groundwater–dominated lowland catchments, which is also reflected in the flat slope of the FDCs (Fig. 3a). The mean BFI ranges of the three catchments (Fig. 7–9) agree well with the previous study by Batelaan and De Smedt (2007), where the range of BFIs are 0.81-0.83 for the Zwarte Beek, 0.79–0.81 for the Demer (the Herk and Mombeek is part of Demer) and 0.81–0.87 for the Dijle, using the Slote and Crouse method (Sloto and Crouse, 1996). In our study, both the Eckhardt and Nathan methods generate a relatively smooth baseflow time series when compared to the graphical HYSEP methods, which still contain large fluctuations in the stream flow likely due to rainfall events and corresponding quick flow (Fig. 7–9). Zomlot et al. (2015) conducted baseflow separation





in 11 selected regional catchments in Flanders and they also found that the one– or two–parameter recursive digital filter methods generated slightly lower mean BFIs than the local minimum method from HYSEP.


Based on the trend and seasonality analysis, we can see that the groundwater discharge to the river is a process where the groundwater (in terms of the representative groundwater level) with strong trend and seasonal signals flows towards the stream flow with weak trend and seasonality (Fig. 23). The trend and seasonality of the separated baseflow lay between these two, with the baseflow from the Nathan method closer to the representative groundwater level direction (Fig. 23). Since there is (1)

a better fit between the simulated baseflow and the separated baseflow from Nathan and Eckhardt methods, (2) a higher correlation coefficient between groundwater level and baseflow from these two methods, and (3) a closer link in trend and seasonality between groundwater level and these two time series, we consider the digital filter Nathan and Eckhardt methods to yield more reliable results than the graphical HYSEP methods in separating baseflow in temperate lowland catchments.

Simulated baseflow time series through impulse response modelling, can capture part of the variation of the separated baseflow, but are not so well in producing the baseflow peaks of the separated baseflow. This may be partly caused by the static drainage level, which is subtracted from the representative groundwater level time series without considering the transient dynamics of drainage, thus makes it difficult to reproduce part of the separated baseflow series. Another influence is from the input time series of the representative groundwater level itself, which has far less peak values if compared with the stream flow

hydrograph. Therefore, it is slightly more difficult to produce the peak baseflow values through impulse response modelling compared to the classic hydrograph separation methods.

Although the mean BFIs of the simulated baseflow time series are relatively lower than the ones from hydrograph separation, our observations show that the impulse response modelling approach from the groundwater flow perspective can be an optional

method to estimate the baseflow, as also suggested by Long (2015). Since the separated baseflow using the traditional graphical or digital filter methods from the stream flow record, might overestimate the contribution of the baseflow to the stream flow, a more smooth baseflow stemmed from the groundwater level, might be a better representation of reality. Moreover, estimation of the baseflow from the groundwater perspective also adds more consideration and weight to the physical processes governing river–aquifer exchange.

**5.2 Impulse response modelling of the hydrological system**

To the best of our knowledge, this is the first study that has used a two–step approach with lumped parameter impulse response modelling for estimating baseflow in a temperate lowland hydrological system.

For the groundwater level response modelling process, the studied lowland aquifers seem to react rather fast to the system

input. A large number of wells reach their peak response during the first day in the three catchments. For the sake of





comparison, we give the response time in groundwater level due to precipitation recharge, estimated in another study (Gonzales et al., 2009) using the impulse response functions by Ventis (Venetic, 1970) and Olsthoorn (Olsthoorn, 2007). A confined coarse sandy aquifer in a Dutch lowland catchment, with an average length of 800 m and an average thickness of 10 m, a hydraulic conductivity of 30 m d$^{-1}$ and a storability of $1.37 \times 10^{-3}$ m m$^{-1}$ (Saliha A. H. and Nonner, 2004), for a precipitation

event of around 8 mm, has a groundwater response delay of approximately 16 h (0.67 d) (Gonzales et al., 2009). This is consistent with the fast response behavior of most wells in our simulations, with peak response during the first day. However, the time step interval of our model is determined by the input data of precipitation and air temperature, which is daily in our case. We thus are not able to capture peak response time shorter than a day. If the model input and level observation time series have higher resolution, the peak response time will probably be less than one day or even just a couple of hours, especially for

the shallow unconfined lowland aquifers.

We explored the potential impacts of well distance to the river and the groundwater depth to the surface on the system impulse response. In the sandy Zwarte Beek catchment, there are no significant impacts observed due to these two factors in the dry period and only a slight increase in the peak time during the wet period for a few wells which are a bit further from the river

and have a bit deeper water tables. Since there is no big difference in the sediment settings of these well locations, the difference in the peak time may be likely caused by high water content during the wet period in the unsaturated zone which prolongs the percolation process. The small, sandy Zwarte Beek catchment shows an overall very fast groundwater level response. In the Herk and Mombeek catchment, we observed the tendency of faster and higher peak response when the well is closer to the river and the groundwater depth shallower to the surface. As the majority of the groundwater level wells, especially from

INBO (Fig. 5b), locate closer to the striped alluvial clay sediment along the river, we also see an impact of the sediment materials on the impulse response. When rainfall events occur, a shallow well close to the river experiences some buffering effect from the clayey sediment in the floodplain close to the river, which delays the process of the activated groundwater to flow fast towards the river. This can lead to a faster and higher response on the near–river groundwater level. On the other hand, when a well is deeper and farther away from the river, it takes a longer time to reach the groundwater table and its

corresponding response magnitude is relatively smaller, and the impact of the near-river hydraulic gradient is also limited. For the Dijle catchment, the individual wells are drilled in a variety of lithologies (e.g. sand, loam, clay and peat), and the wells are clustered closely to the river, so we do not see any peak response link with the distance to the river. However, there is some trend with increase in depth and decrease in response, with the similar reason as in the Herk and Mombeek.

During the second process of baseflow response modelling, the system response is fast and linear. As mentioned before, estimating baseflow from a groundwater perspective can be a promising option. Further research is however recommended to improve the simulation. When extracting the first principle components from the simulated groundwater level time series, more weight can be assigned to wells located close to the river and less weight given to wells afar, for instance, since shallow groundwater closer to the river tends to follow local pathways and brings more contribution to the river. This makes the




generated groundwater level more representative for the near-river status. A second improvement can be made in the model itself. Instead of using a constant drainage level, dynamic drainage level time series could for instance be included by adjusting relevant input codes.

The fitted impulse response models could furthermore potentially be used in future to assess impact of climate change on 690 groundwater level and baseflow, as the only required input data are precipitation and temperature.

**5.3 Time series analysis**

Although there are several observed drought events in Europe in recent years (e.g. 1992, 2003, 2015 and 2018) (Hänsel et al., 2019; Fu et al., 2020), and our annual precipitation time series show low levels for some years in Belgium (e.g. 1996, 2003, 2013 and 2018), the precipitation trend is still small over the observation window of 30 years. Under the impacts of a temperate 695 humid climate, the seasonality of the precipitation has some seasonal variations but not very pronounced (Fig. 2). On the contrary, both the trend and seasonality of the groundwater level time series are pronounced. This is due to the strong seasonal impacts from air temperature, which heavily influences the evapotranspiration process in the unsaturated zone. When the precipitation recharge reaches the shallow aquifer, the groundwater level time series inherits the similar seasonal patterns from air temperature. Thus, we observe strong seasonality of the representative groundwater level time series (Fig. 20-22).


The trend and seasonal strengths of the stream flow lay between the two ends of precipitation and groundwater level (Fig. 23), which agrees with the fact that stream flow is the end-product of the combined influences from groundwater discharge, precipitation and other unidentified flow components. In the Zwarte Beek and Herk and Mombeek, the precipitation (contributing to stream flow as part of the quick flow components) and groundwater level (contributing to stream flow as the 705 slow flow components) have similar weight of impacts on the stream flow, while in the Dijle, the link between precipitation and stream flow seems to be stronger than the link between groundwater level and stream flow (Table 2). From the baseflow response modelling, we know that the groundwater to baseflow process is fast. Therefore, the baseflow tends to carry relatively strong trend and seasonal strengths from the groundwater level. Since the trend and seasonal strengths of the stream flow, compared to the ones of the groundwater level, have already largely reduced (Figure 23), there are indeed some unignorable 710 components, such as interflow and overland flow with less trend and seasonal strengths, contributing significantly to stream flow. This is also in line with our discussion in the sections above, that the graphical and filter methods seem to overestimate the baseflow from the stream flow record, and include part of the interflow and overland flow components within the separated baseflow.



**6 Conclusions**

Through a combined approach of baseflow separation, impulse response modelling and time series analysis, we gained better insights into the river–aquifer interactions and the lowland hydrological system in the three studied catchments.

The graphical HYSEP (fixed interval, sliding interval and local minimum) and recursive digital filter approaches (Nathan and Eckhardt) yield high mean BFIs ($\geq 0.70$), indicating a strong groundwater–dominated feature in the three catchments. Due to

methodological differences, the Nathan and Eckhardt methods generate a relatively smoother baseflow time series than the three graphical HYSEP methods.

We explored the lowland hydrological system with impulse response modelling in two steps, (1) the groundwater level response to system input of precipitation and air temperature, and (2) the baseflow response to system input of groundwater

level. During the first process, groundwater level in shallow aquifers reacts fast to the system input, with most of the wells reaching their peak response during the first day. There is an overall trend of faster response time and higher response magnitude in the wet than the dry periods for the three catchments. The differences in peak time and peak response of individual well can be caused by multiple factors, e.g. well distance to the river, groundwater depth to water table, or local hydrogeological setting. In the sandy Zwarte Beek catchment, the differences in the peak time of a few wells are likely caused

by local effects. In the Herk and Mombeek, the stream bed and bank consist of clay materials, which have some buffering effects for the near–river hydraulic interactions. Thus, there is a tendency of faster peak time and higher peak response when the well is closer to the river and the groundwater depth shallower to the surface. In the Dijle, given a complex sedimentation and river net of the well locations, we do not find strong links between response and distance to the river, while a slightly trend with increase in depth and decrease in response for a few wells.


During the second process, the system response is fast and the simulated baseflow time series tend to be smoother. The simulated time series can capture some variations but not the peaks of the separated baseflow time series. The smooth simulated baseflow leads to lower mean BFIs than the separated ones. They might be a better representation of reality, however, since the separated baseflow directly stemmed from the stream flow record might overestimate the contribution of the baseflow to

the stream flow. The simulated baseflow from the groundwater perspective in contrast considers to some level the physical connection between river and aquifer in the subsurface. The overestimation of baseflow from hydrograph separation is also supported by the time series analysis to some level. The trend and seasonal strengths of the stream flow are not strong, when compared to the ones of the groundwater level and separated baseflow. This indicates that there are other potential flow components such as overland flow and interflow with less trend and seasonal strengths, contributing to the stream flow besides

the groundwater discharge. They are somehow included and estimated as part of the separated baseflow from the hydrograph methods.



The simulated baseflow from the second process showed a better fit with the separated baseflow from Nathan and Eckhardt methods. There is also a higher correlation between groundwater level and baseflow from these two methods, and a closer link in trend and seasonality between groundwater level and these two time series. Therefore, we consider the digital filter Nathan and Eckhardt methods to yield more reliable results than the graphical HYSEP methods in separating baseflow in temperature lowland environments.

## Code availability

The RRAWFLOW code is publicly available at https://www.usgs.gov/centers/dakota-water/science/rrawflow-rainfall-response-aquifer-and-watershed-flow-model?qt-science_center_objects=7#qt-science_center_objects (Long, 2015). R functions from DVstats (Lorenz, 2017), EcoHydRology (DR et al., 2018) and FlowScreen (Dierauer and Whitfield, 2019) packages are available for baseflow separation using HYSEP, Nathan and Eckhardt methods, respectively.

## Data availability

Meteorological input data (precipitation and air temperature) were obtained on request from KMI (Koninklijk Meteorologisch Instituut). Stream flow data are available at https://www.waterinfo.be and downloaded via the wateRinfo R package interface (Van Hoey, 2020). All groundwater data are available via the web services of INBO (Instituut voor Natuur- en Bosonderzoek), DOV (Databank Ondergrond Vlaanderen) and DEE (Département de l'Environnement et de l'Eau).

## Author contribution

ML and BR conceived and designed the study. ML conducted the analyses under the mentorship of BR. ML wrote the manuscript. All authors took part in the discussion of the results and revisions of the manuscript.

## Competing interests

The authors declare that they have no conflict of interest.

## Acknowledgements

The authors would like to thank the editor and two anonymous reviewers for their helpful and insightful comments on the paper.





**Financial support**

Funding for this study was provided by the Fonds Wetenschappelijk Onderzoek in Flanders, Belgium, under grant agreement Nr. S003017N – Future Floodplains: ecosystem services of floodplains under socio-ecological change. The article processing charges for this open-access were covered by KU leuven.

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
