# Peer review of "Exploring river–aquifer interactions and hydrological system response using baseflow separation, impulse response modelling and time series analysis in three temperate lowland catchments"

_Hydrology and Earth System Sciences, 2021_

## Referee Comment (RC2)

Review of "Exploring river-aquifer interactions and hydrological system response….. Lu et al

Franklin Schwartz
The Ohio State University

The paper by Lu et al examines surface water groundwater interactions in lowland catchments in Belgium. The purpose of the study was to fill a gap in knowledge related to these catchments (Line 61). Their approach relied on an impulse response modeling to establish baseflow from knowledge of water table fluctuations. Those baseflow estimates would then be employed to evaluate methods of hydrograph separation and to learn something about the hydrology of these lowland basins.

Study Design

Intuitively, I question the motivations for study, discussed in the introduction. There are many different kinds of watersheds worldwide and it is not clear why the knowledge gap in this case was worthy of the time spent. Another question is the apparent need for another study designed to evaluate the efficacy of various baseflow separation techniques. The paper itself identified the key problem (Line 216) "Limitations for these hydrograph separation methods are their intrinsic difficulty to validate the separated baseflow and the lack of any representation of the physical processes of the river-aquifer exchange". This problem is well known and has been widely explored (e.g., Partington et al., 2012). The positive aspects mentioned in the paper "fast", "efficient", "widely used" and quantitative (line 218) really don't justify techniques know to be little more than guesses in most applications. The choices to address this problem in my opinion are to minimize this aspect of the study in the paper, demonstrate with field data that one of the approaches does work well enough to be useful, or to use a modelling approach like SWOT that might be useful.

The study in my opinion suffers from an over-reliance on theoretically based approaches. On line 48, the paper mentions several field-based approaches, but suggested that these were scale-inappropriate. There are other techniques not mentioned that have been used in other studies, e.g., isotope tracers and geochemical hydrograph separation. These of course come along with their own problems but have been applied to basins of this scale, which are small in area. The paper would be helped by field-based data/observations that could validate any of the empirical conclusions.

The decision to forego a rigorous physically based modeling, approach e.g., HydroGeoSphere, in the study design was surprising. That model was used in various baseflow application e.g., Olsthoorn et al. (2012) – an application looking at the efficacy of hydrograph separation methods (cited in the paper), and with geochemical approaches (Jones et al., 2006). Even if the study was focused on refinement of the impulse-response approach, it would have been prudent to start with a simple well constrained model-based proxy (like Olsthoorn et al. 2012).

What is Baseflow?

I think it is important for the authors to explain their concept of baseflow. The implicit definition in the paper is that stream baseflow is due to groundwater. For developed watersheds, baseflow is flow in the

stream between storm-runoff events. That water could be groundwater, but it also might include slow surface-water discharge from impoundments, storm-water ponds, dewatering, or discharges of treated sewage, etc (Liu et al. 2013). With this expanded definition in mind, the authors need additional field data to support their assumption that baseflow is groundwater.

Development of a watershed (farming, cities etc.) also has the potential to reduce baseflows by decreasing natural groundwater recharge due to tile drains, stormwater collection systems and fast runoff from pavements and altered land cover. To provide context for this study of low land basin these possibilities need to be explored with additional field data and observations.

Questions Concerning the Data

The descriptions of these study watershed appear relatively meager in terms of hydrologic data. First, in looking at the stream hydrographs, it seemed that that discharges were unusually constrained in a narrow range of discharges. I think that for the two smaller basins at least mean daily discharges do not provide adequate temporal resolution of discharge conditions.

In most watersheds, groundwater-level hydrographs are relatively uncommon. The record shown in the paper appears to have combined bits-and-pieces of hydrographs from different wells. But I could be mistaken. The assertion that forcing from precipitation provides a single simulated water-level fluctuation for an entire catchment is a serious simplification that has not really been appropriately justified and is not appropriate. The job of the land-surface component of hydrologic models is to redistribute water on the land surface due to topography, land cover, and soil conditions, which together provide for huge variability in local infiltration rates. Similarly, the hydrologic response of shallower wells could be substantially different than deeper wells because of local variability in hydrogeologic parameters. For example, there are no indication as to whether aquifers are fractured at shallow depth etc.

It is also noticeable that the locations of groundwater observation wells are biased to specific parts of watersheds, and to locations close to streams. Are these completed in alluvial aquifers adjacent to the stream or at what depths? Are these wells special to have water-level records, what kind of records exist etc.?  How often are water levels measured in these wells.

Publication Strategy

With 23 Figures and 45 pages, this paper is overly long with several uncoordinated threads. Yet even at this size, there are major gaps in the description of the hydrogeological setting and data deficiencies that are concerning. My recommendations would be for the authors to rethink their publication strategy to create several papers with different purposes.

With modest effort, there might be a first paper to examine unique features of the hydrologic settings, especially basin morphology, elevation, land use land cover, in predicting hydrographs. It might be necessary to find an approach to reconstruct (downscale) hydrographs to improve resolution. Also, high-resolution water sampling of one storm – with specific conductance etc. together with a few groundwater samples could provide a better understanding of where inter-storm water is coming from.

A second paper might be designed to develop a more sophisticated understanding of water-level behaviors in a model system with a uniform rainfall to begin exploration of the impulse response modeling of the first link – precipitation groundwater.

Finally, a third paper might extend to modeling baseflow as you have done in this paper. But with a much better concept of how everything is working. I would also recommend that you only return to hydrograph separation with a tunable scheme that would integrate basic approaches with some kind of observational approach.

Jones et al 2006  DOI:10.1029/2006WR005416

Liu et al 2013  ISSN : 1866-6280

---

## Author Comment (AC1)

Response to anonymous referee # 1

Dear Anonymous Referee #1,

First of all, thank you very much for reviewing our paper entitled "Exploring river–aquifer interactions and hydrological system response using baseflow separation, impulse response modelling and time series analysis in three temperate lowland catchments (HESS-2021-422)". We are very grateful for your time and comments that help to improve the manuscript.

We have responded point-by-point to your comments and suggestions. Please check the detailed replies in the following sections, with your original comments in *italic* and our answers in blue. A revised manuscript which specifies the adjustments based on your comments will be provided at a later stage, awaiting the review from a second referee and the editor's decision.

We are looking forward to your further assessment.

On behalf of all authors,

With best regards,

Min Lu

Corresponding author

**1. General comments**

*The study looks in details at the hydrological interactions between lowland rivers and shallow aquifers in three different catchments in Belgium. In their analysis, the authors use a combined approach of baseflow separation, impulse response modeling and time series analysis over a 30-year period. Overall, the paper is well-written and the results are insightful and very useful to the hydrology community. I think the paper should be published after addressing the following minor/technical comments:*

- Thank you very much for your positive comments, support for the manuscript and addressing the manuscript's added values to the hydrology research.

**2. Specific suggestions**

*- Looking at Fig. 11(c), I can see a slight overestimation of the groundwater levels prior to 2000. Can the impulse response parameters be tuned to improve the fit to the data? Also the seasons, as described in the text, are not clear on the figure. Consider zooming in.*

- Thank you for the comment and suggestion. Indeed, there is a slight overestimation of the selected groundwater time series P021 in the Dijle catchment (Fig. 11c), especially between 1990 and 2000. If we split the 30-year study period into smaller sub-periods, for example, every 10-year period, and optimize one set of impulse response parameters for each sub-period, the fitting to the observed level time series can be improved.

- However, the option above mentioned deviates a bit from the scope of the study, which assumes that the hydrological system has not evolved drastically within the 30-year time frame (1990 - 2020) and analyzes the hydrological process of groundwater level responses directly to the system input of pure climatic drivers (precipitation and air temperature). Therefore, the model residuals may contain and reflect effects of human interferences, such as increased pumping during some periods or seasons (Fig. 11c). Further, we chose on purpose this not-so-perfect fitted groundwater time series (P021 does not have the highest NSE value among all groundwater level simulations in the Dijle catchment) to indicate that there is indeed some bias during simulation and estimation for some of the groundwater level time series.

- We agree with you that the indication of seasons, especially accompanying the text description of "…(L416) at the end of summer or begin of autumn… " , are not clearly visualized in Fig. 11(c). We improved Fig. 11 by adding minor gridlines on a yearly basis for helping visualizing the low points. We also adjusted the text from "…(L416) at the end of summer or begin of autumn… " to "…(L416) in the middle or second half of the year… ". The updated Fig. 11 is shown below and will be updated in the revised manuscript as well.

[Figure]

**Figure 11** Selected observed (in blue) and simulated (in black) groundwater level time series for impulse response modelling: (a) ZWAP062 in the Zwarte Beek (NSE = 0.838), (b) MOMP012 in the Herk and Mombeek (NSE = 0.857) and (c) DYLP021 in the Dijle (NSE = 0.583).

*- Section 4.2.2: I am interested to look at the entire eigenvalue spectrums. I understand that the first mode at the 3 different sites dominate the rest of the modes. Does the spectrum die after the first leading modes? I expect the leading modes to change if there is some sort of an extreme rainfall event. It might be helpful to comment and discuss this further in the text.*

- Thank you for the comments on the eigenvalue spectrums. We improved Fig. 16 by adding the percentage of variances of the first five principal components (see Fig. 16 below). Figure 16 shows clearly the spectrum dies after the first/leading principle component (Fig. 16b, d, f). The figure will be updated in the revised manuscript as well.

- As this study mainly focuses on an overall catchment behavior for a broader time scale of 30 years in each catchment, we did not assess the impacts on the mode change due to extreme rainfall events, which usually take places at much finer temporal scales such as a few hours or days. The leading modes may change due to extreme rainfall events if checking at finer temporal scales. However, this assessment is out of the research scope and focuses.

[Figure]

**Figure 16** The representative groundwater level (relative elevation) time series and the percentage of variances of the first five principle components in each catchment.

*- Do you expect the BFI estimates in these groundwater-dominated lowland catchments to change if the precipitation regimes change?*

- Thank you for raising the question. Yes, we expect the BFI will change if the precipitation regimes change. For instance, if summers are becoming drier with less precipitation (e.g. dry springs/summers observed between 2017 – 2020 in west Europe), the rivers are at risk of drying up. Expected groundwater discharge to rivers will also be limited due to lowered groundwater level.

- We think that the magnitude of impacts on the BFI due to the precipitation regime change also depend on the temporal scale of the regime change. If the precipitation regime changes over a longer period (e.g. 10 years, or even longer), the impacts on BFI will also demonstration a long-term trend. On the contrary, if the regime change just happens for a few years, its impacts on BFI will be short-term since the river-aquifer interactions in lowland environment are very robust.

*- The figures could be annotated better. For instance, in Fig. 13 does not label the black and the gray curves.*

- Thank you for this suggestion. The IRF curve from each of the retained groundwater level time series is plotted in black (with a transparency setting of 0.3), which makes it grey. Further, the overlaying of some curves darken some plot areas and they look like the black color. To clarify the color misunderstanding, we adjusted the individual curve to black without transparency setting (see Fig. 13 below). The figure will be updated in the revised manuscript as well.

- Following your suggestions, we also checked other figures to improve the annotation and clear visualization. For instances, we will adjust the annotation in Fig. 2 and add minor gridlines in Fig. 12, 17 -19, etc. Detailed figure adjustments and improvement can be found in the tracked and revised manuscript at a later stage.

[Figure]

**Figure 13** IRFs derived from the groundwater level response modelling for dry (April–September) and wet (October–March) periods in the three catchments.

---

## Author Comment (AC2)

Response to Prof. Dr. Franklin Schwartz

Dear Prof. Dr. Franklin Schwartz,

Thank you very much for reviewing our paper entitled "Exploring river–aquifer interactions and hydrological system response using baseflow separation, impulse response modelling and time series analysis in three temperate lowland catchments (HESS-2021-422)". We appreciate your time and comments which will help improving the manuscript.

In the following sections you will find our detailed responses and explanations (in blue) to your comments and suggestions (in *italic*). A revised manuscript will be prepared later with tracked changes made to the original manuscript based on your and other's comments, awaiting the decision from the editor.

We are looking forward to your further assessment.

On behalf of all authors,

With best regards,

Min Lu

Corresponding author

**1. General comments**

*The paper by Lu et al examines surface water groundwater interactions in lowland catchments in Belgium. The purpose of the study was to fill a gap in knowledge related to these catchments (Line 61). Their approach relied on an impulse response modeling to establish baseflow from knowledge of water table fluctuations. Those baseflow estimates would then be employed to evaluate methods of hydrograph separation and to learn something about the hydrology of these lowland basins.*

- Thank you for this comment. As outlined in the introduction section (Line 73 – 74), the first objective of this study is to use the data-driven impulse response modelling to evaluate the hydrological system response (in magnitude and time) to system input of climatic drivers (use case 1; section 3.3.2) or groundwater level (use case 2; section 3.3.3.) in the three temperate lowland catchments. Further, we found that the simulated baseflow from the impulse response modelling (use case 2) can at a certain level provide feedbacks (Line 75 –77) on the separated baseflow from hydrograph separation approaches. To be more precise, the "simulated baseflow" should be adjusted to "simulated groundwater inflows" (to the river). See detailed explanations in the reply to "What is baseflow" part below.

**2. Specific comments**

*Study Design*

*Intuitively, I question the motivations for study, discussed in the introduction. There are many different kinds of watersheds worldwide and it is not clear why the knowledge gap in this case was worthy of the time spent. Another question is the apparent need for another study designed to evaluate the efficacy of various baseflow separation techniques. The paper itself identified the key problem (Line 216) "Limitations for these hydrograph separation methods are their intrinsic difficulty to validate the separated baseflow and the lack of any representation of the physical processes of the river-aquifer exchange". This problem is well known and has been widely explored (e.g., Partington et al., 2012). The positive aspects mentioned in the paper "fast", "efficient", "widely used" and quantitative (line 218) really don't justify techniques know to be little more than guesses in most applications. The choices to address this problem in my opinion are to minimize this aspect of the study in the paper, demonstrate with field data that one of the approaches does work well enough to be useful, or to use a modelling approach like SWOT that might be useful.*

- Thank you for raising the questions and doubts. In the manuscript, we did not mention in detail the study/project background. This study is not a stand-alone one, as it is part of the Future Floodplain project (https://www.futurefloodplains.be/en/) which focuses on the interplay between (geo)hydrology, ecology and geomorphology in the selected catchments. These sites are the focus areas for the research partners and stakeholders.

- We work on the (geo)hydrological package of the project. On the one hand, we need to fill the missing knowledge gaps of river-aquifer interaction in these catchments. On the other hand, we need to provide input for the ecological package of the project, for instance, the characteristic groundwater levels (average highest and lowest groundwater levels). Their ecological modelling requires the input to assess groundwater sensitive species in the floodplains and near the rivers.

- As we mentioned in the response to the general comments, the aim of this study is to evaluate the lowland hydrological system response using a two-step approach (2 use cases) and to explore whether it is potentially feasible to estimate the groundwater contribution to the river from the groundwater level perspective. Although we used the traditional hydrograph separation methods, we did not intend to focus on evaluating methodological differences between varying baseflow separation techniques. We only found that Nathan and Eckhardt methods are more in line with a groundwater-level-based approach in our lowland environment. In the revised manuscript, we will describe this point more clearly.

- Thank you for suggesting the field and modelling approaches. We know that the study of river-aquifer interactions can be very complex. From a broader research perspective, we have divided the river-aquifer interaction study into three main parts: (1) data-driven modelling and simulation at a catchment scale, (2) multi-method field approaches at local scales, and (3) comprehensive numerical modelling approach including both current and future scenarios (climate and land use change). This paper is focused on the first part, and including aspects of the others would make it even more lengthy. Hence we consider this out-of-scope for the current work.

*The study in my opinion suffers from an over-reliance on theoretically based approaches. On line 48, the paper mentions several field-based approaches, but suggested that these were scale-inappropriate. There are other techniques not mentioned that have been used in other studies, e.g., isotope tracers and geochemical hydrograph separation. These of course come along with their own problems but have been applied to basins of this scale, which are small in area. The paper would be helped by field-based data/observations that could validate any of the empirical conclusions.*

- Thank you for raising the concerns about field approaches. We have carried out multi-method field approaches in the focus zones of the catchments (part of the whole catchments). We have collected river water and shallow groundwater samples at different seasons for geohydrochemical (major ions and cations, Radon-222) and isotopic (H2 and O18) analysis. We use heat tracer for estimating river-aquifer interactions at local scales and are currently working on it. We also explored the thermal infrared imagery technique together with Radon-222 analysis in the study sites at local scales (https://onlinelibrary.wiley.com/doi/abs/10.1002/hyp.13839). These are the main focuses of the second part of our river-aquifer interaction studies.

*The decision to forego a rigorous physically based modeling, approach e.g., HydroGeoSphere, in the study design was surprising. That model was used in various baseflow application e.g., Olsthoorn et al. (2012) – an application looking at the efficacy of hydrograph separation methods (cited in the paper), and with geochemical approaches (Jones et al., 2006). Even if the study was focused on refinement of the impulse-response approach, it would have been prudent to start with a simple well constrained model-based proxy (like Olsthoorn et al. 2012).*

- Thank you for pointing out the numerical modelling approach. We did not forego it. Numerical modelling of river-aquifer interactions is one of the most important approaches for us to learn and understand the river-aquifer interaction of the hydrological system. This is the main focus of the third part of our project. Based on our previous modelling experience, we chose MODFLOW-based numerical modelling. We are currently working on it and will compare the groundwater inflow to the river from numerical MODFLOW approach and data-driven modelling approach (of this manuscript).

*What is Baseflow?*

*I think it is important for the authors to explain their concept of baseflow. The implicit definition in the paper is that stream baseflow is due to groundwater. For developed watersheds, baseflow is flow in the*

*stream between storm-runoff events. That water could be groundwater, but it also might include slow surface-water discharge from impoundments, storm-water ponds, dewatering, or discharges of treated sewage, etc (Liu et al. 2013). With this expanded definition in mind, the authors need additional field data to support their assumption that baseflow is groundwater.*

*Development of a watershed (farming, cities etc.) also has the potential to reduce baseflows by decreasing natural groundwater recharge due to tile drains, stormwater collection systems and fast runoff from pavements and altered land cover. To provide context for this study of low land basin these possibilities need to be explored with additional field data and observations.*

- Thank you for the comments on the baseflow. We agree it is necessary to explicitly define the concept of baseflow in our study. In the revised manuscript, we will define baseflow from the hydrograph separation as (total) baseflow, which includes the groundwater inflow to the river (groundwater contribution), and other intermediate water that sustains streamflow between rainfall events. The "simulated baseflow" will be adjusted to "simulated groundwater inflow" to the river. Section 3.3.3 will be groundwater inflow modelling instead of baseflow response modelling. We will also adjust the corresponding terms throughout the whole manuscript where necessary, and hence adopt the more nuanced definition of baseflow, as suggested.

- We have checked the paper by Liu et al. (2013). The studied streams are located in an urban setting (average impervious coverage ~ 50%). The baseflow is mainly composed of groundwater inflow and water released from the storm-water detention ponds (Liu et al. 2013). Unlike the urban streams influenced by anthropogenic activities, the three catchments of this study are dominated by crop and meadow, and have relatively low urban coverage (Line 129 – 131). Zwarte Beek (Dirk Maes, 1992) and Mombeek valleys are nature reserves with low human impacts. The Belgium government has carried out nature-based solution for floodplain restoration and "zero management" polices to reduce the human impacts on catchments such as Dijle since 1990 (Turkelboom et al, 2021). Therefore, the three catchments are mostly under the natural conditions, which makes it feasible to implement the impulse-response modelling, especially under climatic drivers. We will add more description regards to the natural state of the catchments in the revised manuscript.

*Questions Concerning the Data*

*The descriptions of these study watershed appear relatively meager in terms of hydrologic data. First, in looking at the stream hydrographs, it seemed that that discharges were unusually constrained in a narrow range of discharges. I think that for the two smaller basins at least mean daily discharges do not provide adequate temporal resolution of discharge conditions.*

- Thank you for raising this question. We described the hydrologic data such as precipitation, stream flow in the study area section, providing the average values or the value range, with some graphical representation of the data (Fig. 2 and 3). We will add some more statistical description of the data

in the revised manuscript. We chose daily discharge values over finer resolution (e.g. hourly) because we simulated the impulse response modelling over a relative long period (30 years) and focused on the temporal evolution at a coarse temporal resolution. Using daily values for all three catchments helped us to compare the difference between them. For event based or finer temporal resolution studies, we agree it is better to have much finer temporal scales than daily.

*In most watersheds, groundwater-level hydrographs are relatively uncommon. The record shown in the paper appears to have combined bits-and-pieces of hydrographs from different wells. But I could be mistaken. The assertion that forcing from precipitation provides a single simulated water-level fluctuation for an entire catchment is a serious simplification that has not really been appropriately justified and is not appropriate. The job of the land-surface component of hydrologic models is to redistribute water on the land surface due to topography, land cover, and soil conditions, which together provide for huge variability in local infiltration rates. Similarly, the hydrologic response of shallower wells could be substantially different than deeper wells because of local variability in hydrogeologic parameters. For example, there are no indication as to whether aquifers are fractured at shallow depth etc.*

- Thank you for raising the question about the groundwater-level hydrographs. It is common to quantify baseflow from a streamflow hydrograph perspective. We explored in this study the possibility to get a sense of groundwater contribution to the river from a groundwater-level hydrograph perspective, which is not so common but also can be considered as the innovative part of our study.

- As mentioned above, the three catchments have experienced little human impact during the study period. The aim of the study is to use a simple data-driven model to explore the hydrological system response under climatic drivers before applying a complicated numerical modelling including other influencing factors.

- Since most groundwater level observations are close to the river, we used the first principal component to get a collective representation of the relative state of the shallow groundwater levels across the catchment for facilitating the application of the impulse response model for the second use case (Line 328 – 331). The groundwater level observations in this study are from shallow depths so they represent the water table elevation, and fractures are not of concern in these shallow unconsolidated sediments. Locally, Roer-Valley Graben related faults influence groundwater levels in the region, but in the current study areas, there is no evidence they would reach up to the shallow aquifer.

*It is also noticeable that the locations of groundwater observation wells are biased to specific parts of watersheds, and to locations close to streams. Are these completed in alluvial aquifers adjacent to the stream or at what depths? Are these wells special to have water-level records, what kind of records exist etc.? How often are water levels measured in these wells.*

- Thank you for raising the questions. We tried our best to get as many groundwater level observations as possible from different monitoring agencies (e.g. Flanders and Wallonia have different monitoring networks and schemes). Since our focus is on exploring the river-aquifer interactions, the well locations close to the streams actually can provide more valuable information for the regional groundwater inflow to rivers. We have recommended that weighting

scheme reflecting this might improve the approach further (Line 682 – 685), but consider it out of scope in the current work.

- In the second paragraph of the section "3.1.3 Data cleaning", we described the groundwater level observations (Line 167-178). In general, groundwater level observations from different sources have different measurement intervals, which can be daily, weekly, biweekly or monthly. They are all from shallow aquifers, with the depth to water table less than 20m. We will improve the description of groundwater level observations in this part to be more clear.

*Publication Strategy*

*With 23 Figures and 45 pages, this paper is overly long with several uncoordinated threads. Yet even at this size, there are major gaps in the description of the hydrogeological setting and data deficiencies that are concerning. My recommendations would be for the authors to rethink their publication strategy to create several papers with different purposes.*

*With modest effort, there might be a first paper to examine unique features of the hydrologic settings, especially basin morphology, elevation, land use land cover, in predicting hydrographs. It might be necessary to find an approach to reconstruct (downscale) hydrographs to improve resolution. Also, high-resolution water sampling of one storm – with specific conductance etc. together with a few groundwater samples could provide a better understanding of where inter-storm water is coming from.*

*A second paper might be designed to develop a more sophisticated understanding of water-level behaviors in a model system with a uniform rainfall to begin exploration of the impulse response modeling of the first link – precipitation groundwater.*

*Finally, a third paper might extend to modeling baseflow as you have done in this paper. But with a much better concept of how everything is working. I would also recommend that you only return to hydrograph separation with a tunable scheme that would integrate basic approaches with some kind of observational approach.*

*Jones et al 2006 DOI:10.1029/2006WR005416*

*Liu et al 2013 ISSN : 1866-6280*

- Thank you for the suggestions. As this study is focused on the river and shallow aquifer interactions, we included a general description of the hydrogeological setting and focused on describing the surficial geology. We will improve the hydrogeological setting description in the study area section in the revised manuscript.

- We agree that the paper is long. Most papers cover only one study site, while in this study three catchments are included. The data preparation and method description are also described quite in detail. Considering your suggestions, we would like to shorten the manuscript a bit, for instance, reduce the description of the data preparation section, shift some figures (Figure 6, 10) to appendix or remove them if they are deviated from the main objectives, and leave out some less relevant descriptions in the original manuscript.

- We appreciate the suggested publication strategy of three papers. However, this work already is part of a larger strategy, with the field data interpretation and numerical modelling approaches being next. Therefore, we would prefer to address the length of the manuscript, and focus more on the main goal instead of splitting it up.

References

- Dirk, Maes. The use of monitoring systems in nature reserves, an example: "De Vallei van de Zwarte Beek" at Koersel-Beringen (Limburg, Belgium). Conference: 8th International Colloquium of the European Invertebrate Survey , 1992.
- Turkelboom, F., Demeyer, R., Vranken, L. et al. How does a nature-based solution for flood control compare to a technical solution? Case study evidence from Belgium. Ambio 50, 1431–1445, https://doi.org/10.1007/s13280-021-01548-4, 2021.

---

## Author Response (AR1)

Author's response

Dear Prof. Dr. Nadia Ursino,

We thank you for the opportunity to improve our manuscript entitled "Exploring river–aquifer interactions and hydrological system response using baseflow separation, impulse response modelling and time series analysis in three temperate lowland catchments (HESS-2021-422)". We have addressed all the comments from the two referees in the revised manuscript.

We have reduced the length of the original manuscript from 45 to 36 pages and listed the major changes in the revised manuscript below.

**Major revisions:**

- Improved the description of the hydrologic data, lithological and hydrogeological settings, the natural state of the catchments in the "Study area" section
- Integrated the previous "Data collection" and "Data cleaning" sections into "Data collection" and added detailed description of the groundwater level observations
- Shortened the "Data imputation", "Model overview" of "Impulse response modelling", and removed the "Cross correlation analysis" in the "Methodology" section
- Removed previous Fig. 5, 6, 10, 12, 15, Table 2 and the relevant text in the original manuscript
- Improved visualization and annotations of all figures, and avoided the verbal explanations in the captions, e.g. Fig.2, Fig.5–7, Fig. 12–17
- Adjusted the "Results", "Discussion" and "Conclusions" sections according to the methodology, figure changes as listed above

In the following sections, we have listed the detailed changes (in blue) made to the original manuscript based on the comments (in *italic*). The page and line numbers (e.g. **P2, L26**) correspond to the revised manuscript (see the supplement).

We are looking forward to the further assessment.

On behalf of all authors,

With best regards,

Min Lu

Corresponding author

Response to Anonymous Referee #1

**1. General comments**

*The study looks in details at the hydrological interactions between lowland rivers and shallow aquifers in three different catchments in Belgium. In their analysis, the authors use a combined approach of baseflow separation, impulse response modeling and time series analysis over a 30-year period. Overall, the paper is well-written and the results are insightful and very useful to the hydrology community. I think the paper should be published after addressing the following minor/technical comments:*

- We thank the Anonymous Referee #1 for his/her useful comments and suggestions to help improve the quality of the manuscript. We have addressed all the comments in the revised manuscript, see the detailed responses to specific suggestions below.

**2. Specific comments**

*- Looking at Fig. 11(c), I can see a slight overestimation of the groundwater levels prior to 2000. Can the impulse response parameters be tuned to improve the fit to the data? Also the seasons, as described in the text, are not clear on the figure. Consider zooming in.*

- As mentioned in the previous response to Anonymous Referee #1, if we split the 30-year study period into smaller sub-periods, for example, every 10-year period, and optimize one set of impulse response parameters for each sub-period, the fitting to the observed level time series could be improved. But this approach deviates from the scope of this study, which assumes that the hydrological system has not evolved drastically within the whole 30-year time frame (1990 - 2020) and uses one set of impulse response parameters covering the whole period. The model residuals and deviations from the simulated time series may reflect effects of human interferences.

- We agree with you that Fig. 8 (originally Fig. 11) and the previously described text did not match well. We improved Fig. 8 **(P16)** by adding minor gridlines on a yearly basis for helping visualize the low points, and adjusted the text from "…exhibit very low levels at the end of summer or begin of autumn…" to "…exhibit very low levels in the middle or second half of the year…" **(P16, L349)** for clarification.

*- Section 4.2.2: I am interested to look at the entire eigenvalue spectrums. I understand that the first mode at the 3 different sites dominate the rest of the modes. Does the spectrum die after the first leading modes? I expect the leading modes to change if there is some sort of an extreme rainfall event. It might be helpful to comment and discuss this further in the text.*

- Yes, the spectrum dies after the first mode. To make this point clear, we improved Fig. 11 **(P20)** by adding the percentage of variances of the first five principal components in each catchment. Figure 11 shows clearly the trend of spectrum signals (Fig. 11b, d, f). Since this study mainly focuses on an overall catchment behavior for a broader time scale of 30 years, we did not assess the impacts on the mode change due to extreme rainfall events, which usually take places at much finer temporal scales such as a few hours or days. The leading modes may change due to extreme rainfall events if checking at finer temporal scales. However, this assessment is out of the research scope and focuses.

*- Do you expect the BFI estimates in these groundwater-dominated lowland catchments to change if the precipitation regimes change?*

- Yes, we expect the BFI will change if the precipitation regimes change. For instance, if summers are becoming drier with less precipitation (e.g. dry springs/summers observed between 2017 – 2020 in western Europe), the rivers are at risk of drying up. Expected groundwater discharge to rivers will also be limited due to lowered groundwater level.

- We think that the magnitude of impacts on the BFI due to the precipitation regime change also depends on the temporal scale of the regime change. If the precipitation regime changes over a longer period (e.g. 10 years, or even longer), the impacts on BFI will also demonstrate a long-term trend. On the contrary, if the regime change just happens for a few years, its impacts on BFI will be short-term since the river-aquifer interactions in lowland environment are very robust.

*- The figures could be annotated better. For instance, in Fig. 13 does not label the black and the gray curves.*

- We agree with the suggestion. We improved Fig. 9 (originally Fig. 13) by plotting the IRF curves all in black and removed the previous transparency setting **(P17)**. In this way, there is no confusion about black/grey color differences. Following your suggestions, we also improved annotations and visualization of all other figures, and avoided the verbal explanations in the captions. For instance, we adjusted the left y axis and the colors for presenting air temperature curves in Fig. 2 **(P4)**. Now it is easier to distinguish the air temperature and precipitation without specifying the plot types in the caption. We also added minor gridlines or annotations in Fig. 5–7 **(P13–15)** and Fig. 12–17 **(P22–27)** for better visualization. Other detailed adjustments on figures can be found in the revised manuscript with tracked changes.

Response to Prof. Dr. Franklin Schwartz

**1. General comments**

*The paper by Lu et al examines surface water groundwater interactions in lowland catchments in Belgium. The purpose of the study was to fill a gap in knowledge related to these catchments (Line 61). Their approach relied on an impulse response modeling to establish baseflow from knowledge of water table fluctuations. Those baseflow estimates would then be employed to evaluate methods of hydrograph separation and to learn something about the hydrology of these lowland basins.*

- We appreciate the helpful comments and suggestions for improving our manuscript. We revised the manuscript and addressed all the comments, see the detailed point-to-point responses below.

**2. Specific comments**

*Study Design*

*Intuitively, I question the motivations for study, discussed in the introduction. There are many different kinds of watersheds worldwide and it is not clear why the knowledge gap in this case was worthy of*

*the time spent. Another question is the apparent need for another study designed to evaluate the efficacy of various baseflow separation techniques. The paper itself identified the key problem (Line 216) "Limitations for these hydrograph separation methods are their intrinsic difficulty to validate the separated baseflow and the lack of any representation of the physical processes of the river-aquifer exchange". This problem is well known and has been widely explored (e.g., Partington et al., 2012). The positive aspects mentioned in the paper "fast", "efficient", "widely used" and quantitative (line 218) really don't justify techniques know to be little more than guesses in most applications. The choices to address this problem in my opinion are to minimize this aspect of the study in the paper, demonstrate with field data that one of the approaches does work well enough to be useful, or to use a modelling approach like SWOT that might be useful.*

- As mentioned in the previous response to Prof. Dr. Franklin Schwartz, the selected sites are of interest for multiple research partners and stakeholders of the Future Floodplain Project. Studying the river-aquifer interaction can fill the missing gap and also provide input for the ecological studies in these catchments. We made this point clear in the introduction section **(P2–3, L54–59)** .

- We adjusted slightly the phrasing of the research objectives **(P3, L65–69)**. The first objective is to simulate the groundwater level response using impulse response modelling, and the second one is to simulate the groundwater inflow to rivers and compare the estimated groundwater inflow with the separated baseflow. Although we used the traditional hydrograph separation methods, we did not intend to focus on evaluating methodological differences between varying baseflow separation techniques. We removed the description such as "…we consider the digital filter Nathan and Eckhardt methods to yield more reliable results than the graphical HYSEP methods…" from the original manuscript to avoid misleading or shifting of the main research focus.

- From a broader research perspective, we have divided the river-aquifer interaction study into three main parts: (1) data-driven modelling and simulation at a catchment scale, (2) multi-method field approaches at local scales, and (3) comprehensive numerical modelling approach including both current and future scenarios (climate and land use change). This paper focuses on the first part, and including aspects of the others would make it even more lengthy.

*The study in my opinion suffers from an over-reliance on theoretically based approaches. On line 48, the paper mentions several field-based approaches, but suggested that these were scale-inappropriate. There are other techniques not mentioned that have been used in other studies, e.g., isotope tracers and geochemical hydrograph separation. These of course come along with their own problems but have been applied to basins of this scale, which are small in area. The paper would be helped by field-based data/observations that could validate any of the empirical conclusions.*

- We have carried out multi-method field approaches in the focus zones of the catchments (part of the whole catchments). We have collected river water and shallow groundwater samples at different seasons for geohydrochemical (major ions and cations, Radon-222) and isotopic (H2 and O18) analysis. We use heat tracer for estimating river-aquifer interactions at local scales and are currently working on it. We also explored the thermal infrared imagery technique together with Radon-222 analysis in the study sites at local scales (https://onlinelibrary.wiley.com/doi/abs/10.1002/hyp.13839). These are the main focuses of the second part of our river-aquifer interaction studies.

*The decision to forego a rigorous physically based modeling, approach e.g., HydroGeoSphere, in the study design was surprising. That model was used in various baseflow application e.g., Olsthoorn et al. (2012) – an application looking at the efficacy of hydrograph separation methods (cited in the paper), and with geochemical approaches (Jones et al., 2006). Even if the study was focused on refinement of the impulse-response approach, it would have been prudent to start with a simple well constrained model-based proxy (like Olsthoorn et al. 2012).*

- As part of a broader research scheme, numerical modelling of river-aquifer interactions is the main focus of the third subsection of our project. Based on our previous modelling experience, we chose MODFLOW-based numerical modelling. We are currently working on it and will compare the groundwater inflow to the river from numerical MODFLOW approach and data-driven modelling approach (of this manuscript).

*What is Baseflow?*

*I think it is important for the authors to explain their concept of baseflow. The implicit definition in the paper is that stream baseflow is due to groundwater. For developed watersheds, baseflow is flow in the*

*stream between storm-runoff events. That water could be groundwater, but it also might include slow surface-water discharge from impoundments, storm-water ponds, dewatering, or discharges of treated sewage, etc (Liu et al. 2013). With this expanded definition in mind, the authors need additional field data to support their assumption that baseflow is groundwater.*

*Development of a watershed (farming, cities etc.) also has the potential to reduce baseflows by decreasing natural groundwater recharge due to tile drains, stormwater collection systems and fast runoff from pavements and altered land cover. To provide context for this study of low land basin these possibilities need to be explored with additional field data and observations.*

- We agree it is necessary to explicitly define the concept of baseflow in our study. In the revised manuscript, we defined "baseflow, from groundwater discharge and other delayed sources" **(P8, L177–178)**. The term "baseflow response modelling" was adjusted to "groundwater inflow response modelling" **(P11, L270)** as groundwater inflow to rivers is part of the (total) baseflow. We also adjusted the corresponding terms throughout the whole manuscript where necessary to distinguish between baseflow and groundwater inflow.

- Unlike urban streams influenced by anthropogenic activities (Liu et al. 2013), the three catchments of this study are dominated by crop and meadow, and have relatively low urban coverage **(P6, L122–124)**. Human impact on these catchments was also limited during the research period. Therefore, the three catchments reflect natural conditions, which makes it feasible to implement the impulse-response modelling, especially under climatic forcing. We added more descriptions with regard to the natural state of the catchments **(P6, L124–128)**.

*Questions Concerning the Data*

*The descriptions of these study watershed appear relatively meager in terms of hydrologic data. First, in looking at the stream hydrographs, it seemed that that discharges were unusually constrained in a*

*narrow range of discharges. I think that for the two smaller basins at least mean daily discharges do not provide adequate temporal resolution of discharge conditions.*

- Besides the graphical presentation of the hydrologic data (Fig. 2 and 3), we added and revised the statistical description part **(P3, L76–82)**. Regarding discharge, we chose daily discharge values over finer resolution (e.g. hourly) because we simulated the impulse response modelling over a relatively long period (30 years) and focused on the temporal evolution at a coarse temporal resolution. Also, using daily values for all three catchments helped us to compare the difference between them. For event based or finer temporal resolution studies, we agree it is better to have much finer temporal scales than daily.

*In most watersheds, groundwater-level hydrographs are relatively uncommon. The record shown in the paper appears to have combined bits-and-pieces of hydrographs from different wells. But I could be mistaken. The assertion that forcing from precipitation provides a single simulated water-level fluctuation for an entire catchment is a serious simplification that has not really been appropriately justified and is not appropriate. The job of the land-surface component of hydrologic models is to redistribute water on the land surface due to topography, land cover, and soil conditions, which together provide for huge variability in local infiltration rates. Similarly, the hydrologic response of shallower wells could be substantially different than deeper wells because of local variability in hydrogeologic parameters. For example, there are no indication as to whether aquifers are fractured at shallow depth etc.*

- We explored in this study the possibility to get a sense of groundwater contribution to the river from a groundwater-level hydrograph perspective, which is not so common but also can be considered as the innovative part of our study. Since the three catchments have experienced little human impact during the study period, this makes it possible to use a simple data-driven model to explore the hydrological system response under climatic forcing first, before applying a complicated numerical modelling including other influencing factors.

- As most groundwater level observations are close to the river, we used the first principal component to get a collective representation of the relative state of the shallow groundwater levels across the catchment for facilitating the application of the impulse response model for the second use case **(P11, L272–275)**. The groundwater level observations in this study are from shallow depths so they represent the water table elevation, and fractures are not of concern in these shallow unconsolidated sediments. Locally, Roer-Valley Graben related faults influence groundwater levels in the region, but in the current study areas, there is no evidence they would reach up to the shallow aquifer.

*It is also noticeable that the locations of groundwater observation wells are biased to specific parts of watersheds, and to locations close to streams. Are these completed in alluvial aquifers adjacent to the stream or at what depths? Are these wells special to have water-level records, what kind of records exist etc.? How often are water levels measured in these wells.*

- We tried our best to get as many groundwater level observations as possible from different monitoring agencies (e.g. Flanders and Wallonia have different monitoring networks and schemes). Since our focus is on exploring the river-aquifer interactions, the well locations close to the streams actually can provide more valuable information for the regional groundwater inflow to rivers. We

have recommended that weighting scheme reflecting this might improve the approach further **(P29, L522–525)**, but consider it out of scope in the current work.

- In the original manuscript, the description of the groundwater level time series wasa bit scattered and not specific. We have improved and added more detailed description in the data collection section. We specified the sources of observations, record intervals, time series lengths, groundwater depths in the three catchments **(P7, L149–164)**.

*Publication Strategy*

*With 23 Figures and 45 pages, this paper is overly long with several uncoordinated threads. Yet even at this size, there are major gaps in the description of the hydrogeological setting and data deficiencies that are concerning. My recommendations would be for the authors to rethink their publication strategy to create several papers with different purposes.*

*With modest effort, there might be a first paper to examine unique features of the hydrologic settings, especially basin morphology, elevation, land use land cover, in predicting hydrographs. It might be necessary to find an approach to reconstruct (downscale) hydrographs to improve resolution. Also, high-resolution water sampling of one storm – with specific conductance etc. together with a few groundwater samples could provide a better understanding of where inter-storm water is coming from.*

*A second paper might be designed to develop a more sophisticated understanding of water-level behaviors in a model system with a uniform rainfall to begin exploration of the impulse response modeling of the first link – precipitation groundwater.*

*Finally, a third paper might extend to modeling baseflow as you have done in this paper. But with a much better concept of how everything is working. I would also recommend that you only return to hydrograph separation with a tunable scheme that would integrate basic approaches with some kind of observational approach.*

*Jones et al 2006 DOI:10.1029/2006WR005416*

*Liu et al 2013 ISSN : 1866-6280*

- We added additional descriptions on the lithostratigraphic classifications in the subsurface **(P5, L108–110, L112–115)**. Since this study is focused on the river and shallow aquifer interactions, we gave a bit more weight on describing the surficial geology.

- We agree that the paper is long. We have shortened the manuscript from 45 to 36 pages. For instance, we have integrated the previous Sect 3.1.2 to Sect 3.1.1 Data collection, removed previous Fig. 5, 6, 10, 12, 15, Table 2 and the relevant text, etc. Other detailed adjustments can be traced in the revised manuscript.

- We appreciate the suggested publication strategy of three papers. However, this work already is part of a larger strategy, with the field data interpretation and numerical modelling approaches being next. Therefore, we would prefer to focus more on the main goal instead of splitting it up.

---

## Author Response (AR2)

Response to editor-in-chief

Dear Prof. Dr. Nadia Ursino,

Thank you very much for the feedback and the decision of considering accepting our manuscript after minor revision. We have revised our manuscript with point-to-point responses to the comments from anonymous referee # 3, and also addressed the question of the specific target of our study. The revised files, reply to comments and marked-up manuscript are uploaded.

We are looking forward to your further revision.

On behalf of all authors,

With best regards,

Min Lu

Corresponding author

Response to anonymous referee # 3

Dear Anonymous Referee # 3,

Thank you very much for reviewing our paper and we are very grateful for your time and comments that help to improve the manuscript.

We have responded point-by-point to your comments and suggestions. Please check the detailed replies in the following sections, with your original comments in *italic* and our answers in blue. A revised manuscript which specifies the adjustments based on your comments is also provided in an attachment. The line numbers below refer to the revised manuscript.

We are looking forward to your further assessment.

On behalf of all authors,

With best regards,

Min Lu

Corresponding author

**1. General comments**

*This manuscript describes a study on three watersheds in low-lying areas of Belgium. The key interest is to describe how groundwater levels respond to weather (precipitation and temperature—a stand-in for evaporation rate), and seasonality, and then how groundwater influences or contributes to streamflow.*

*Various techniques are used to separate baseflow from runoff, and impulse-response models are used to model the links from precipitation to groundwater levels to baseflow. The impulse-response methods used have separate terms for slow and fast responses. In addition, time-series analysis is used to determine long-term trends and seasonality in key variables.*

*I found the methods used and the analysis to be interesting and worthwhile. However, the manuscript seems to be ambivalent about its purpose. That is, are the authors interested in these three watersheds, or in all similar watersheds? Are they looking for the techniques that work for these watersheds, or are they providing a specific example that can be followed for other watersheds? My impression is that it is a combination—specific techniques applied to characterize specific watersheds that can be used to characterize other watersheds as well. If this last combination is the intention of the authors, it would be helpful if they would make it more apparent.*

*The basic structure of the project—comparing results from baseflow separation to results from the impulse-response modeling—was a clever combination of empirical and modeling approaches that yielded important characteristics of the watersheds.*

- Thank you very much for your positive comments, support for the manuscript and addressing the novelty of the combined approach used in this study.

- Regards to the purpose of this study, your impression is right. It is indeed with a broader vision in potential applications. We wanted to develop a combined approach to reveal the close links between different components (precipitation – groundwater – baseflow – streamflow) in the lowland hydrological system, without initiating a complicated distributed model at the first step. By applying and testing the combined approach in three specific catchments, we would like to know whether it can work well in these catchments and whether it can be further applied in all similar catchments or not.

- We thank you for this suggestion and agree to make this point more apparent. In the introduction part, we added in Line 69 and 70 "If this combined approach can be successfully applied in these three specific catchments, it can potentially be applied to catchments with similar conditions worldwide.". In the conclusion part, we recalled back to this point by adding the sentence in Line 578 and 579 "Based the performances of the combined approach in our study sites, we consider this approach has further potential to be applied to similar lowland catchments with small area coverages and under natural conditions or limited human impacts.".

**2. Specific comments**

*- Line 73—two watershed areas are given but there are three watersheds.*

- Thank you for the comment. After checking this line, we think there is a misunderstanding of the decimal separator. In this sentence, comma is not used as a decimal separator but an element separator for digits 95 and 272, which are the areas of the first two catchments.

*- Figure 11—Perhaps an intuitive explanation for the principal component time series?*

- Thank you for the suggestion. We agree that it is necessary to explain more clearly the principal component time series. We added in Line 398 from "… to a relative elevation (in meter) …" to "… to a relative elevation (in meter, dividing the scores of the first principle component by the sum of the loadings) …" and a new sentence in Line 399-401 "This time series can be interpreted intuitively as the relative difference of the groundwater level across the catchment at a certain point-in-time, with the average groundwater level.".